# Not All Neuro-Symbolic Concepts Are Created Equal: Analysis and Mitigation of Reasoning Shortcuts

**Emanuele Marconato**
DISI and DI
University of Trento and University of Pisa
Trento, Italy
emanuele.marconato@unitn.it

**Stefano Teso**
CIMeC and DISI
University of Trento
Trento, Italy
stefano.teso@unitn.it

**Antonio Vergari**
School of Informatics
University of Edinburgh
Edinburgh, UK
avergari@exseed.ed.ac.uk

**Andrea Passerini**
DISI
University of Trento
Trento, Italy
andrea.passerini@unitn.it

## Abstract

Neuro-Symbolic (NeSy) predictive models hold the promise of improved compliance with given constraints, systematic generalization, and interpretability, as they allow to infer labels that are consistent with some prior knowledge by reasoning over high-level concepts extracted from sub-symbolic inputs. It was recently shown that NeSy predictors are affected by *reasoning shortcuts*: they can attain high accuracy but by leveraging concepts with *unintended semantics*, thus coming short of their promised advantages. Yet, a systematic characterization of reasoning shortcuts and of potential mitigation strategies is missing. This work fills this gap by characterizing them as unintended optima of the learning objective and identifying four key conditions behind their occurrence. Based on this, we derive several natural mitigation strategies, and analyze their efficacy both theoretically and empirically. Our analysis shows reasoning shortcuts are difficult to deal with, casting doubts on the trustworthiness and interpretability of existing NeSy solutions.

## 1  Introduction

Neuro-Symbolic (NeSy) AI aims at improving the *robustness* and *trustworthiness* of neural networks by integrating them with reasoning capabilities and prior knowledge [1, 2]. We focus on *NeSy predictors* [3, 4], neural structured-output classifiers that infer one or more labels by reasoning over high-level *concepts* extracted from sub-symbolic inputs, like images or text [5, 6, 7, 8, 9, 10, 11]. They leverage reasoning techniques to encourage – or even *guarantee* – that their predictions comply with domain-specific regulations and inference rules. As such, they hold the promise of improved *systematic generalization*, *modularity*, and *interpretability*, in that learned concepts can be readily reused in different NeSy tasks, as done for verification [12], and for explaining the model's inference process to stakeholders [13, 14, 15]. On paper, this makes NeSy predictors ideal for high-stakes applications that require both transparency and fine-grained control over the model's (in- and out-of-distribution) behavior, such as medical diagnosis [16], robotics [17] and self-driving cars [18].

Much of the promise of these models relies on learned concepts being *high quality*. The general consensus is that the prior knowledge constrains learned concepts to behave as expected [19] and issues with them are often tackled heuristically [20]. It was recently shown that, however, NeSy predictors can *attain high accuracy by leveraging concepts with unintended semantics* [21]. Following

37th Conference on Neural Information Processing Systems (NeurIPS 2023).

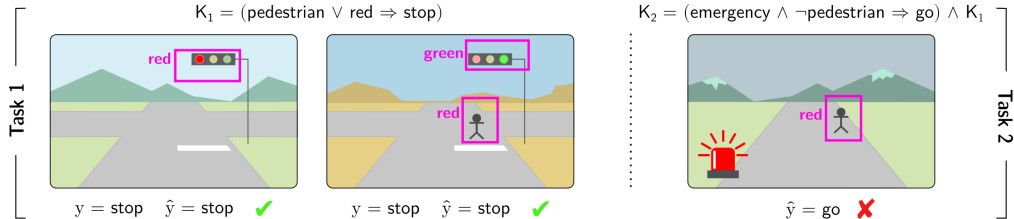

Figure 1: **Reasoning shortcuts undermine trustworthiness**. An autonomous vehicle has to decide whether to $Y = \texttt{stop}$ or $Y = \texttt{go}$ based on three binary concepts extracted from an image $\mathbf{x}$, namely $C_1 = \texttt{red}$ light, $C_2 = \texttt{green}$ light and $C_3 = \text{presence of } \texttt{pedestrians}$ (shown in **pink**). **Left**: In Task 1, the prior knowledge $\mathsf{K} = (\texttt{pedestrian} \lor \texttt{red} \Rightarrow \texttt{stop})$ instructs the vehicle to stop whenever the light is red or there are pedestrians on the road. The model can perfectly classify an (even exhaustive) training set by acquiring a *reasoning shortcut that classifies pedestrians as red lights*. **Right**: The learned concepts are then reused to guide an autonomous ambulance with the additional rule that in emergency situations red lights can be ignored, with potentially dire consequences. Our work identifies the causes of RSs (Section 4) and several mitigation strategies (Section 5).

Marconato et al. [21], we refer to these as *reasoning shortcuts* (RSs). RSs are problematic, as concepts encoding unintended semantics compromise generalization across NeSy tasks, as shown in Fig. 1, as well as interpretability and verification of NeSy systems [12]. Moreover, the only known mitigation strategies are based on heuristics [20, 11].

The issue is that RSs – and their root causes – are not well understood, making it difficult to design effective remedies. In this paper, we fill this gap. We introduce a formal definition of RSs and theoretically characterize their properties, highlighting how they are a general phenomenon affecting a variety of state-of-the-art NeSy predictors. Specifically, our results show that RSs can be shared across many NeSy architectures, and provide a way of *counting* them for any given learning problem. They also show that RSs depend on *four key factors*, namely the structure of the prior knowledge and of the data, the learning objective, and the architecture of the neural concept extractor. This enables us to identify several supervised and unsupervised mitigation strategies, which we systematically analyze both theoretically and empirically. Finally, we experimentally validate our findings by testing a number of representative NeSy predictors and mitigation strategies on four NeSy data sets.

**Contributions.** Summarizing, we: (*i*) Formalize RSs and identify four key root causes. (*ii*) Show that RSs are a general issue impacting a variety of NeSy predictors. (*iii*) Identify a number of mitigation strategies and analyze their effectiveness, or lack thereof. (*iv*) Empirically show that RSs arise even when the data set is large and unbiased, and evaluate the efficacy of different mitigation strategies, highlighting the limits of unsupervised remedies and the lack of a widely applicable recipe.

## 2 The Family of Neuro-Symbolic Predictors

**Notation.** Throughout, we indicate scalar constants $x$ in lower-case, random variables $X$ in upper case, and ordered sets of constants $\mathbf{x}$ and random variables $\mathbf{X}$ in bold typeface. Also, $\mathbf{x}_{i:j}$ denotes the subset $\{x_i, \ldots, x_j\}$, $[n]$ the set $\{1, \ldots, n\}$, and $\mathbf{x} \models \mathsf{K}$ indicates that $\mathbf{x}$ satisfies a logical formula $\mathsf{K}$.

**NeSy predictors.** A *NeSy predictor* is a model that infers $n$ *labels* $\mathbf{Y}$ (taking values in $\mathcal{Y}$) by reasoning over a set of $k$ discrete *concepts* $\mathbf{C}$ (taking values in $\mathcal{C}$) extracted from a sub-symbolic continous *input* $\mathbf{X}$ (taking values in $\mathcal{X}$). Reasoning can be implemented in different ways, but overall its role is to encourage the model's predictions to comply with given *prior knowledge* $\mathsf{K}$, in the sense that predictions $\mathbf{y}$ that do not satisfy the knowledge are generally avoided. The prior knowledge $\mathsf{K}$ is assumed to be provided upfront and correct, as formalized in Section 3. Normally, only supervision on the labels $\mathbf{Y}$ is available for training, with the concepts $\mathbf{C}$ treated as latent variables.

**Example 1.** *In* MNIST-Addition *[7], given a pair of MNIST images [22], say* $\mathbf{x} = (\boxed{2}, \boxed{6})$*, the model has to infer the concepts* $\mathbf{C} = (C_1, C_2)$ *encoding the digit classes, to predict their sum* $Y$*, in this case* $8$*. Reasoning drives the model towards complying with constraint* $\mathsf{K} = (Y = C_1 + C_2)$*.*

The concepts are modeled by a conditional distribution $p_\theta(\mathbf{C} \mid \mathbf{X})$ parameterized by $\theta \in \Theta$, typically implemented with a neural network. The predicted concepts can be viewed as "soft" or "neural" predicates with a truth value ranging in $[0, 1]$. As for the reasoning step, the most popular strategies

involve *penalizing* the model for producing concepts and/or labels inconsistent with the knowledge at training time [8, 23, 24] or introducing a *reasoning layer* that infers labels from the predicted concepts and also operates at inference time [7, 9, 25, 10]. In either case, end-to-end training requires to differentiate through the knowledge. Mainstream options include softening the knowledge using fuzzy logic [26, 6, 27, 11] and casting reasoning in terms of probabilistic logics [28, 7, 10].

To investigate the scope and impact of RSs, we consider three representative NeSy predictors. The first one is **DeepProbLog** (DPL) [7], which implements a sound probabilistic-logic reasoning layer on top of the neural predicates. DPL is a discriminative predictor of the form [21]:

$$p_\theta(\mathbf{y} \mid \mathbf{x}; \mathsf{K}) = \sum_{\mathbf{c}} u_{\mathsf{K}}(\mathbf{y} \mid \mathbf{c}) \cdot p_\theta(\mathbf{c} \mid \mathbf{x}) \tag{1}$$

where the concept distribution is fully factorized and the label distribution is *uniform*[1] over all label-concept combinations compatible with the knowledge, that is, $u_{\mathsf{K}}(\mathbf{y} \mid \mathbf{c}) = \mathbb{1}\{\mathbf{c} \models \mathsf{K}[\mathbf{Y}/\mathbf{y}]\}/Z(\mathbf{c}; \mathsf{K})$ where the indicator $\mathbb{1}\{\mathbf{c} \models \mathsf{K}[\mathbf{Y}/\mathbf{y}]\}$ *guarantees* all labels inconsistent with K have zero probability, and $Z(\mathbf{c}; \mathsf{K}) = \sum_{\mathbf{y}} \mathbb{1}\{\mathbf{c} \models \mathsf{K}[\mathbf{Y}/\mathbf{y}]\}$ is a normalizing constant. Inference amounts to computing a most likely label $\mathrm{argmax}_{\mathbf{y}} \, p_\theta(\mathbf{y} \mid \mathbf{x}; \mathsf{K})$, while learning is carried out via maximum (log-)likelihood estimation, that is given a training set $\mathcal{D} = \{(\mathbf{x}, \mathbf{y})\}$, maximizing

$$\mathcal{L}(p_\theta, \mathcal{D}, \mathsf{K}) := \tfrac{1}{|\mathcal{D}|} \sum_{(\mathbf{x},\mathbf{y}) \in \mathcal{D}} \log p_\theta(\mathbf{y} \mid \mathbf{x}; \mathsf{K}). \tag{2}$$

In general, it is intractable to evaluate Eq. (1) and solve inference exactly. DPL leverages knowledge compilation [29, 30] to make both steps practical.

Our analysis on DPL can be carried over to other NeSy predictors implementing analogous reasoning layers [31, 32, 33, 10, 34]. Furthermore, we show that certain RSs affect also alternative NeSy approaches such as the **Semantic Loss** (SL) [8] and **Logic Tensor Networks** (LTNs) [6], two state-of-the-art penalty-based approaches. Both reward a neural network for predicting labels $\mathbf{y}$ consistent with the knowledge, but SL measures consistency in probabilistic terms, while LTNs use fuzzy logic to measure a fuzzy degree of knowledge satisfaction. See Appendix A for a full description.

# 3   Reasoning Shortcuts as Unintended Optima

It was recently shown that NeSy predictors are vulnerable to *reasoning shortcuts* (RSs), whereby the model attains high accuracy by leveraging concepts with *unintended semantics* [21].

**Example 2.** *To build intuition, consider* MNIST-Addition *and assume the model is trained on examples of only two sums:* 🄋 + 🄊 $= 1$ *and* 🄋 + 🄋 $= 2$. *Note that there exist two distinct maps from images to concepts that perfectly classify such examples: one is the intended solution (*🄋 $\mapsto 0$, 🄊 $\mapsto 1$, 🄋 $\mapsto 2$*), while the other is (*🄋 $\mapsto 1$, 🄊 $\mapsto 0$, 🄋 $\mapsto 1$*). The latter is unintended.*

**The ground-truth data generating process.** In order to properly define what a RS is, we have to first define what *non*-shortcut solutions are. In line with work on causal representation learning [35, 36, 37], we do so by specifying the ground-truth data generation process $p^*(\mathbf{X}, \mathbf{Y}; \mathsf{K})$. Specifically, we assume it takes the form illustrated in Fig. 2. In short, we assume there exist $k$ unobserved ground-truth concepts $\mathbf{G}$ (*e.g.*, in MNIST-Addition these are the digits 0 to 9) that determine *both* the observations $\mathbf{X}$ (the MNIST images) and the labels $\mathbf{Y}$ (the sum). We also allow for extra stylistic factors $\mathbf{S}$, independent from $\mathbf{G}$, that do influence the observed data (*e.g.*, calligraphic style) but *not* the labels. The ground-truth concepts are discrete and range in $\mathcal{G} = [m_1] \times \cdots \times [m_k]$, whereas the style $\mathbf{s} \in \mathbb{R}^q$ is continuous. The training and test examples $(\mathbf{x}, \mathbf{y})$ are then obtained by first sampling $\mathbf{g}$ and $\mathbf{s}$ and then $\mathbf{x} \sim p^*(\mathbf{X} \mid \mathbf{g}, \mathbf{s})$ and $\mathbf{y} \sim p^*(\mathbf{Y} \mid \mathbf{g}; \mathsf{K})$. Later on, we will use $\mathrm{supp}(\mathbf{G})$ to denote the support

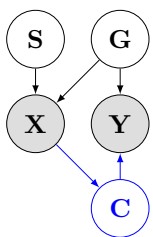

Figure 2: The ground-truth data generation process (in **black**) and a NeSy predictor (in **blue**).

of $p^*(\mathbf{G})$. We also assume that the ground-truth process is consistent with the prior knowledge K, in the sense that invalid examples are never generated: $p^*(\mathbf{y} \mid \mathbf{g}; \mathsf{K}) = 0$ for all $(\mathbf{g}, \mathbf{y})$ that violate K.

**What is a reasoning shortcut?** By definition, a NeSy predictor $p_\theta(\mathbf{Y} \mid \mathbf{X}; \mathsf{K})$ – shown in **blue** in Fig. 2 – acquires concepts with the correct semantics if it recovers the ground-truth concepts, *i.e.*,

$$p_\theta(\mathbf{C} \mid \mathbf{x}) \equiv p^*(\mathbf{G} \mid \mathbf{x}) \quad \forall \mathbf{x} \in \mathcal{X} \tag{3}$$

---

[1]In practice, the distribution of labels given concepts needs not be uniform [10].

A model satisfying Eq. (3) easily generalizes to other NeSy prediction tasks that make use of the same ground-truth concepts $\mathbf{G}$, as $p_\theta(\mathbf{C} \mid \mathbf{X})$ can be reused for solving the new task [21, 38]. It is also interpretable, in the sense that as long as stakeholders understand the factors $\mathbf{G}$, they can also interpret concept-based explanations of the inference process that rely on $\mathbf{C}$ [39]. Naturally, there may exist many concept distributions that *violate* Eq. (3) and that as such do *not* capture the correct semantics. However, all those that achieve sub-par log-likelihoods can be ideally avoided simply by improving learning or supplying more examples. We define RSs as those concept distributions for which these strategies are not enough.

**Definition 1.** *A reasoning shortcut is a distribution $p_\theta(\mathbf{C} \mid \mathbf{X})$ that achieves maximal log-likelihood on the training set but does not match the ground-truth concept distribution,*

$$\mathcal{L}(p_\theta, \mathcal{D}, \mathsf{K}) = \max_{\theta' \in \Theta} \ \mathcal{L}(p_{\theta'}, \mathcal{D}, \mathsf{K}) \qquad \wedge \qquad p_\theta(\mathbf{C} \mid \mathbf{X}) \not\equiv p^*(\mathbf{G} \mid \mathbf{X}) \tag{4}$$

This makes RSs difficult to improve by regular means and also hard to spot based on predictions alone. Yet, since the concepts do not recover the correct semantics, RSs can compromise systematic generalization and interpretability. For instance, the shortcut concepts learned in Example 2 would fail completely at `MNIST-Addition` tasks involving digits other than $0$, $1$, and $2$ and also at other arithmetic tasks, like multiplication, involving the same digits, as we show experimentally in Section 6. An examples of RSs in a high-stakes scenario is shown in Fig. 1.

## 4 Properties of Reasoning Shortcuts

**Counting deterministic RSs.** We begin by assessing how many RSs exist in an idealized setting in which the ground-truth generative process is simple, we have access to the true risk, and the concepts $\mathbf{C}$ have been specified correctly, that is, $\mathcal{C} = \mathcal{G}$, and then proceed to work out what this entails. Specifically, we work with these assumptions:

**A1**. The distribution $p^*(\mathbf{X} \mid \mathbf{G}, \mathbf{S})$ is induced by a map $f : (\mathbf{g}, \mathbf{s}) \mapsto \mathbf{x}$, *i.e.*, $p^*(\mathbf{X} \mid \mathbf{G}, \mathbf{S}) = \delta\{\mathbf{X} = f(\mathbf{g}, \mathbf{s})\}$, where $f$ is *invertible*, and *smooth* over $\mathbf{s}$.

**A2**. The distribution $p^*(\mathbf{Y} \mid \mathbf{G}; \mathsf{K})$ is induced by a map $\beta_\mathsf{K} : \mathbf{g} \mapsto \mathbf{y}$. This is what happens in `MNIST-Addition`, as there exists a unique value $y$ that is the sum of any two digits $(g_1, g_2)$.

Our analysis builds on a link between the NeSy predictor $p_\theta(\mathbf{Y} \mid \mathbf{X}; \mathsf{K})$ and the concept extractor $p_\theta(\mathbf{C} \mid \mathbf{X})$, which depend on $\mathbf{X}$, and their analogues $p_\theta(\mathbf{Y} \mid \mathbf{G}; \mathsf{K})$ and $p_\theta(\mathbf{C} \mid \mathbf{G})$ that depend directly on the ground-truth concepts $\mathbf{G}$. This link is formalized by the following lemma:

**Lemma 1.** *It holds that: (i) The true risk of $p_\theta$ can be upper bounded as follows:*

$$\mathbb{E}_{(\mathbf{x}, \mathbf{y}) \sim p^*(\mathbf{X}, \mathbf{Y}; \mathsf{K})}[\log p_\theta(\mathbf{y} \mid \mathbf{x}; \mathsf{K})] \le \mathbb{E}_{\mathbf{g} \sim p(\mathbf{G})}\big( - \mathsf{KL}[p^*(\mathbf{Y} \mid \mathbf{g}; \mathsf{K}) \parallel p_\theta(\mathbf{Y} \mid \mathbf{g}; \mathsf{K})] - \mathsf{H}[p^*(\mathbf{Y} \mid \mathbf{g}; \mathsf{K})]\big) \tag{5}$$

*where $\mathsf{KL}$ is the Kullback-Leibler divergence and $\mathsf{H}$ is the Shannon entropy. Moreover, under A1 and A2, $p_\theta(\mathbf{Y} \mid \mathbf{X}; \mathsf{K})$ is an optimum of the LHS of Eq. (5) if and only if $p_\theta(\mathbf{Y} \mid \mathbf{G}; \mathsf{K})$ is an optimum of the RHS. (ii) Under A1, there exists a bijection between the deterministic concept distributions $p_\theta(\mathbf{C} \mid \mathbf{X})$ that are constant over the support of $p(\mathbf{X} \mid \mathbf{g})$, for each $\mathbf{g} \in \mathrm{supp}(\mathbf{G})$, and the deterministic distributions of the form $p_\theta(\mathbf{C} \mid \mathbf{G})$.*

All proofs can be found in Appendix B. Lemma 1 implies that the deterministic concept distributions $p_\theta(\mathbf{C} \mid \mathbf{X})$ of NeSy predictors $p_\theta(\mathbf{Y} \mid \mathbf{X}; \mathsf{K})$ that maximize the LHS of Eq. (5), including those that are RSs, correspond one-to-one to the deterministic distributions $p_\theta(\mathbf{C} \mid \mathbf{G})$ yielding label distributions $p_\theta(\mathbf{Y} \mid \mathbf{G}; \mathsf{K})$ that maximize the RHS of Eq. (5). Hence, we can count the number of *deterministic* RSs by counting the deterministic distributions $p_\theta(\mathbf{C} \mid \mathbf{G})$:

**Theorem 2.** *Let $\mathcal{A}$ be the set of mappings $\alpha : \mathbf{g} \mapsto \mathbf{c}$ induced by all possible deterministic distributions $p_\theta(\mathbf{C} \mid \mathbf{G})$, i.e., each $p_\theta(\mathbf{C} \mid \mathbf{G}) = \mathbb{1}\{\mathbf{C} = \alpha(\mathbf{G})\}$ for exactly one $\alpha \in \mathcal{A}$. Under A1 and A2, the number of deterministic optima $p_\theta(\mathbf{C} \mid \mathbf{G})$ of Eq. (5) is:*

$$\sum_{\alpha \in \mathcal{A}} \mathbb{1}\Big\{\bigwedge_{\mathbf{g} \in \mathsf{supp}(\mathbf{G})} (\beta_\mathsf{K} \circ \alpha)(\mathbf{g}) = \beta_\mathsf{K}(\mathbf{g})\Big\} \tag{6}$$

Intuitively, this sum counts the deterministic concept distributions $p_\theta(\mathbf{C} \mid \mathbf{X})$ – embodied here by the maps $\alpha$ – that output concepts predicting a *correct* label for each example in the training set. The

Table 1: **Impact of different mitigation strategies on the number of deterministic optima**: R is reconstruction, C supervision on $\mathbf{C}$, MTL multi-task learning, and DIS disentanglement. All strategies reduce the number of $\alpha$'s in Eq. (6), sometimes substantially, but require different amounts of effort to be put in place. Actual counts for our data sets are reported in Appendix C.2.

| MITIGATION | REQUIRES | CONSTRAINT ON $\alpha$ | ASSUMPTIONS | RESULT |
|---|---|---|---|---|
| None | – | $\bigwedge_{\mathbf{g}\in\text{supp}(\mathbf{G})} \left((\beta_\mathsf{K} \circ \alpha)(\mathbf{g}) = \beta_\mathsf{K}(\mathbf{g})\right)$ | **A1**, **A2** | Theorem 2 |
| MTL | Tasks | $\bigwedge_{\mathbf{g}\in\text{supp}(\mathbf{G})} \bigwedge_{t\in[T]} \left((\beta_{\mathsf{K}^{(t)}} \circ \alpha)(\mathbf{g}) = \beta_{\mathsf{K}^{(t)}}(\mathbf{g})\right)$ | **A1**, **A2** | Proposition 4 |
| C | Sup. on $\mathbf{C}$ | $\bigwedge_{\mathbf{g}\in\mathcal{S}\subseteq\text{supp}(\mathbf{G})} \bigwedge_{i\in I} \left(\alpha_i(\mathbf{g}) = g_i\right)$ | **A1** | Proposition 5 |
| R | – | $\bigwedge_{\mathbf{g},\mathbf{g}'\in\text{supp}(\mathbf{G}):\mathbf{g}\neq\mathbf{g}'} \left(\alpha(\mathbf{g}) \neq \alpha(\mathbf{g}')\right)$ | **A1**, **A3** | Proposition 6 |

ground-truth distribution $p^*(\mathbf{G} \mid \mathbf{X})$ is one such distribution, so the count is always at least one, but there may be more, and all of these are RSs. Eq. (6) gives us their exact number. This formalizes the intuition of Marconato et al. [21] that, as long as the prior knowledge K admits the correct label $\mathbf{y}$ to be inferred from more than one concept vector $\mathbf{c}$, there is room for RSs. So far, we have assumed Eq. (2) is computed as in DPL. However, RSs are chiefly a property of the *prior knowledge*, and as such also affect NeSy predictors employing different reasoning procedures or different relaxations of the knowledge. We show this formally in Appendix A. Deterministic RS are also important because – in certain cases – they define a basis for *all* reasoning shortcuts:

**Proposition 3.** *For probabilistic logic approaches (including DPL and SL): (i) All convex combinations of two or more deterministic optima $p_\theta(\mathbf{C} \mid \mathbf{X})$ of the likelihood are also (non-deterministic) optima. However, not all convex combinations can be expressed in DPL and SL. (ii) Under A1 and A2, all optima of the likelihood can be expressed as a convex combination of deterministic optima. (iii) If A2 does not hold, there may exist non-deterministic optima that are not convex combinations of deterministic ones. These may be the only optima.*

Combining Proposition 3 (*i*) with Theorem 2 gives us a lower bound for the number of *non-deterministic* RSs, in the sense that if there are at least two deterministic RS, then there exist infinitely many non-deterministic ones. An important consequence is that, if we can somehow control what deterministic RSs affect the model, then we may be able to implicitly lower the number of *non-deterministic* RSs as well. However, Proposition 3 implies that there may exist *non-deterministic* RSs that are unrelated to the deterministic ones and that as such cannot be controlled this way.

## 5 Analysis of Mitigation Strategies

The key factors underlying the occurrence of deterministic RSs appear explicitly in Eq. (6). These are: (*i*) the knowledge K, (*ii*) the structure of $\text{supp}(\mathbf{G})$, (*iii*) the objective function $\mathcal{L}$ used for training (via Lemma 1), and (*iv*) the architecture of the concept extractor $p_\theta(\mathbf{C} \mid \mathbf{X})$, embodied in the Theorem by $p_\theta(\mathbf{C} \mid \mathbf{G})$. This gives us a starting point for identifying possible mitigation strategies and analyzing their impact on the number of *deterministic* RSs. Our main results are summarized in Table 1.

### 5.1 Knowledge-based Mitigation

The *prior knowledge* K is the main factor behind RSs and also a prime target for mitigation. The most direct way of eliminating unintended concepts is to edit K directly, for instance by eliciting additional constraints from a domain expert. However, depending on the application, this may not be feasible: experts may not be available, or it may be impossible to constrain K without also eliminating concepts with the intended semantics.

A more practical alternative is to employ *Multi-Task Learning* (MTL). The idea is to train a NeSy predictor over $T$ tasks sharing the same ground-truth concepts $\mathbf{G}$ but differing prior knowledge $\mathsf{K}^{(t)}$, for $t \in [T]$. *E.g.*, one could learn a model to predict both the sum and product of MNIST digits, as in our experiments (Section 6). Intuitively, by constraining the concepts to work well across tasks, MTL leaves less room for unintended semantics. The following result confirms this intuition:

**Proposition 4.** *Consider $T$ NeSy prediction tasks with knowledge $\mathsf{K}^{(t)}$, for $t \in [T]$ and data sets $\mathcal{D}^{(t)}$, all sharing the same $p^*(\mathbf{G})$. Under A1 and A2, any deterministic optimum $p_\theta(\mathbf{C} \mid \mathbf{G})$ of the MTL loss (i.e., the average of per-task losses) is a deterministic optimum of a single task with prior*

*knowledge $\bigwedge_t \mathsf{K}^{(t)}$. The number of deterministic optima amounts to:*

$$\sum_{\alpha \in \mathcal{A}} \mathbb{1}\left\{ \bigwedge_{\mathbf{g} \in \mathsf{supp}(\mathbf{G})} \bigwedge_{t=1}^{T} \left( (\beta_{\mathsf{K}^{(t)}} \circ \alpha)(\mathbf{g}) = \beta_{\mathsf{K}^{(t)}}(\mathbf{g}) \right) \right\} \tag{7}$$

This means that, essentially, MTL behaves like a logical conjunction: any concept extractor $p_\theta(\mathbf{C} \mid \mathbf{G})$ incompatible with the knowledge of *any* task $t$ is not optimal. This strategy can be very effective, and indeed it performs very well in our experiments, but it necessitates gathering or designing a *set* of correlated learning tasks, which may be impractical in some situations.

## 5.2 Data-based Mitigation

Another key factor is the support of $p^*(\mathbf{G})$: if the support is not full, the conjunction in Eq. (6) becomes looser, and the number of $\alpha$'s satisfying it increases. This is what happens in MNIST-EvenOdd (Example 2): here, RSs arise precisely because the training set only includes a *subset* of combinations of digits, leaving ample room for acquiring unintended concepts.

We stress, however, that RSs can also occur if the data set is *exhaustive*, as in the next example.

**Example 3** (XOR task). *Consider a task with three binary ground-truth concepts $\mathbf{G} = (G_1, G_2, G_3)$ in which the label $Y$ is the parity of these bits, that is $\mathsf{K} = (Y = G_1 \oplus G_2 \oplus G_3)$. Each label $Y \in \{0, 1\}$ can be inferred from four possible concept vectors $\mathbf{g}$, meaning that knowing $\mathbf{y}$ is not sufficient to identify the $\mathbf{g}$ it was generated from. In this case, it is impossible to pin down the ground-truth distribution $p^*(\mathbf{C} \mid \mathbf{G}; \mathsf{K})$ even if all possible combinations of inputs $\mathbf{x}$ are observed.*

One way of avoiding RSs is to explicitly guide the model towards satisfying the condition in Eq. (3) by supplying *supervision* for a subset of concepts $\mathbf{C}_I \subseteq \mathbf{C}$, with $I \subseteq [k]$, and then augmenting the log-likelihood with a cross-entropy loss over the concepts of the form $\sum_{i \in I} \log p_\theta(C_i = g_i \mid \mathbf{x})$. Here, training examples $(\mathbf{x}, \mathbf{g}_I, \mathbf{y})$ come with annotations for the concepts indexed by $I$. The impact of this strategy on the number of deterministic RSs is given by the following result:

**Proposition 5.** *Assume that concept supervision is available for all $\mathbf{g}$ in $\mathcal{S} \subseteq \mathsf{supp}(\mathbf{G})$. Under **A1**, the number of deterministic optima $p_\theta(\mathbf{C} \mid \mathbf{G})$ minimizing the cross-entropy over the concepts is:*

$$\sum_{\alpha \in \mathcal{A}} \mathbb{1}\left\{ \bigwedge_{\mathbf{g} \in \mathcal{S}} \bigwedge_{i \in I} \alpha_i(\mathbf{g}) = g_i \right\} \tag{8}$$

This strategy is very powerful: if $I = [k]$, $\mathcal{S} \equiv \mathsf{supp}(\mathbf{G})$, and the support is complete, there exists only *one* map $\alpha$ that is consistent with the condition in Eq. (8) and it is the identity. Naturally, this comes at the cost of obtaining dense annotations for all examples, which is often impractical.

## 5.3 Objective-based Mitigation

A natural alternative is to augment the log-likelihood with an *unsupervised* penalty designed to improve concept quality. We focus on reconstruction penalties like those used in auto-encoders [40, 41, 42, 43, 44], which encourage the model to capture all information necessary to reconstruct the input $\mathbf{x}$. To see why these might be useful, consider Example 2. Here, the model learns a RS mapping both ◩ and ◪ to the digit 1: this RS hinders reconstruction of the input images, and therefore could be avoided by introducing a reconstruction penalty.

In order to implement this, we introduce additional latent variables $\mathbf{Z}$ that capture the style $\mathbf{S}$ of the input $\mathbf{X}$ and modify the concept extractor to output both $\mathbf{C}$ and $\mathbf{Z}$, that is: $p_\theta(\mathbf{c}, \mathbf{z} \mid \mathbf{x}) = p_\theta(\mathbf{c} \mid \mathbf{x}) \cdot p_\theta(\mathbf{z} \mid \mathbf{x})$. The auto-encoder reconstruction penalty is then given by:

$$\mathcal{R}(\mathbf{x}) = -\mathbb{E}_{(\mathbf{c}, \mathbf{z}) \sim p_\theta(\mathbf{c}, \mathbf{z} \mid \mathbf{x})}\left[ \log p_\psi(\mathbf{x} \mid \mathbf{c}, \mathbf{z}) \right] \tag{9}$$

where, $p_\psi(\mathbf{x} \mid \mathbf{c}, \mathbf{z})$ is the decoder network. We need to introduce an additional assumption **A3**: the encoder and the decoder separate content from style, that is, $p_\theta(\mathbf{C}, \mathbf{Z} \mid \mathbf{G}, \mathbf{S}) := \mathbb{E}_{\mathbf{x} \sim p^*(\mathbf{x} \mid \mathbf{G}, \mathbf{S})} p_\theta(\mathbf{C}, \mathbf{Z} \mid \mathbf{x})$ factorizes as $p_\theta(\mathbf{C} \mid \mathbf{G}) p_\theta(\mathbf{Z} \mid \mathbf{S})$ and $p_\psi(\mathbf{G}, \mathbf{S} \mid \mathbf{C}, \mathbf{Z}) := \mathbb{E}_{\mathbf{x} \sim p_\psi(\mathbf{x} \mid \mathbf{C}, \mathbf{Z})} p^*(\mathbf{G}, \mathbf{Z} \mid \mathbf{x})$ as $p_\psi(\mathbf{G} \mid \mathbf{C}) p_\psi(\mathbf{S} \mid \mathbf{Z})$. In this case, we have the following result:

**Proposition 6.** *Under **A1** and **A3**, the number of deterministic distributions $p_\theta(\mathbf{C} \mid \mathbf{G})$ that minimize the reconstruction penalty in Eq. (9) is:*

$$\sum_{\alpha \in \mathcal{A}} \mathbb{1}\left\{ \bigwedge_{\mathbf{g}, \mathbf{g}' \in \mathsf{supp}(\mathbf{G}): \mathbf{g} \neq \mathbf{g}'} \alpha(\mathbf{g}) \neq \alpha(\mathbf{g}') \right\} \tag{10}$$

| | XOR | | | MNIST-Addition | | |
|---|---|---|---|---|---|---|
| | DPL | SL | LTN | DPL | SL | LTN |
| − | 100% | 100% | 100% | 96.7% | 82.9% | 100% |
| DIS | 0% | 0% | 0% | 0% | 0% | 0% |

Table 2: **Q1: Disentanglement (DIS) can be very powerful** to lower the frequency of RSs on XOR and MNIST-Addition data sets (the lower the better). Results are averaged over 30 *optimal* runs.

In words, this shows that indeed optimizing for reconstruction facilitates disambiguating between different concepts, *i.e.*, different ground-truth concepts cannot be mapped to the same concept. However, minimizing the reconstruction can be non-trivial in practice, especially for complex inputs.

### 5.4 Architecture-based Mitigation

One last factor is the *architecture of the concept extractor* $p_\theta(\mathbf{C} \mid \mathbf{X})$, as it implicitly controls the number of candidate deterministic maps $\mathcal{A}$ and therefore the sum in Theorem 2. If the architecture is unrestricted, $p_\theta(\mathbf{C} \mid \mathbf{X})$ can in principle map any ground-truth concept $\mathbf{g}$ that generated $\mathbf{x}$ to any concept $\mathbf{c}$, thus the cardinality of $\mathcal{A}$ increases exponentially with $k$.

A powerful strategy for reducing the size of $\mathcal{A}$ is *disentanglement*. A model is disentangled if and only if $p_\theta(\mathbf{C} \mid \mathbf{G})$ factorizes as $\prod_{j \in [k]} p_\theta(C_j \mid G_j)$ [45, 36]. In this case, the maps $\alpha$ also factorize into per-concept maps $\alpha_j : [m_j] \to [m_j]$, dramatically reducing the cardinality of $\mathcal{A}$, as shown by our first experiment. In applications where the $k$ concepts are naturally independent from one another, *e.g.*, digits in MNIST-Addition, one can implement disentanglement by predicting each concept using the same neural network, although more general techniques exist [46, 47].

### 5.5 Other Heuristics based on Entropy Regularization

Besides these mitigation strategies, we investigate empirically the effect of the Shannon entropy loss, defined as $1 - \frac{1}{k} \sum_{i=1}^{k} \mathsf{H}_{m_i}[p_\theta(c_i)]$, which was shown to increase concept quality in DPL [20]. Here, $p_\theta(\mathbf{C})$ is the marginal distribution over the concepts, and $\mathsf{H}_{m_i}$ is the normalized Shannon entropy over $m_i$ possible values for the distribution. Notice that this term goes to zero only when each distribution $p_\theta(C_i)$ is uniform, which may conflict with the real objective of the NeSy prediction (especially when only few concepts are observed). Other similar heuristics that are suited for reducing over-confidence in label predictions are based on label smoothing [48], energy-based models [49], annealing [50], and many others [51, 52, 53]. In principle, when applied at the concept level, they help reduce the over-confidence in concepts that is typical in deterministic RSs. While they could also be beneficial to mildly reduce the number of RSs, we take the Shannon entropy regularization as a representative of this family for our experiments.

## 6 Case Studies

In this section, we evaluate the impact of RSs in synthetic and real-world NeSy prediction tasks and how the mitigation strategies discussed in Section 5 fare in practice. More details about the models and data are reported in Appendix C. The **code** is available at github.com/reasoning-shortcuts.

**Q1: More data does not prevent RSs but disentanglement helps.** We start by evaluating the robustness of DPL, SL, and LTN to RSs in two settings where the data set is *exhaustive*. In XOR (cf. Example 3), the goal is to predict the parity of three binary concepts $\mathbf{g} \in \{0, 1\}^3$ given prior knowledge $\mathsf{K} = (y = g_1 \oplus g_2 \oplus g_3)$. The predictor receives the ground-truth concepts $\mathbf{g}$ as input and has to learn a distribution $p_\theta(\mathbf{C} \mid \mathbf{G})$. In MNIST-Addition (cf. Example 1) the goal is to correctly predict the sum of two MNIST digits, *e.g.*, $\mathbf{x} = (\mathit{1}, \mathit{2})$ and $y = 3$. In both tasks, the training set contains examples of *all* possible combinations of concepts, *i.e.*, $\mathsf{supp}(\mathbf{G}) = \mathcal{G}$.

Since the tasks are relatively simple, we can afford to study models that achieve near-optimal likelihood, for which Definition 1 approximately applies. To this end, for each NeSy architecture, we train several models using different seeds and stop as soon as we obtain 30 models with likelihood $\geq 0.95$. Then, we measure the percentage of models that have acquired a RS. We do the same also for models modified to ensure they are disentangled (DIS in the Table), see Appendix C for details. The results, reported in Table 2, clearly show that RSs affect *all* methods if disentanglement is not in place. This confirms that, without implicit architectural biases, *optimizing for label accuracy alone is not*

Table 3: **Q2: Impact of mitigation strategies.** We report the $F_1$-score on the labels (**Y**) and concepts (**C**). All tested methods incorporate DIS and are averaged over 10 runs. **Top**: NeSy methods combined with R, C, H on `MNIST-EvenOdd`. **Bottom**: evaluation on single tasks *vs* MTL on `MNIST-AddMul`.

| MNIST-EvenOdd | DPL | | SL | | LTN | |
|---|---|---|---|---|---|---|
| | $F_1$ (**Y**) | $F_1$ (**C**) | $F_1$ (**Y**) | $F_1$(**C**) | $F_1$ (**Y**) | $F_1$(**C**) |
| – | 85.1 ± 4.6 | 0.1 ± 0.1 | 99.3 ± 0.2 | 0.2 ± 0.1 | 98.1 ± 0.2 | 0.3 ± 0.1 |
| R | 79.8 ± 1.0 | 0.1 ± 0.0 | 99.5 ± 0.2 | 0.1 ± 0.0 | 76.3 ± 1.1 | 0.0 ± 0.0 |
| H | 98.1 ± 0.1 | 0.1 ± 0.1 | 99.4 ± 0.1 | 0.1 ± 0.0 | 81.9 ± 0.5 | 53.9 ± 0.7 |
| C | 84.9 ± 0.1 | 0.1 ± 0.1 | 99.3 ± 0.4 | 21.5 ± 6.2 | 98.1 ± 0.2 | 0.2 ± 0.1 |
| R + H | 75.4 ± 0.4 | 0.2 ± 0.1 | 99.6 ± 0.1 | 0.1 ± 0.0 | 97.9 ± 2.3 | 38.1 ± 16.7 |
| R + C | 84.0 ± 2.2 | 1.9 ± 4.4 | 99.3 ± 0.2 | 61.5 ± 7.8 | 98.1 ± 0.2 | 0.2 ± 0.1 |
| H + C | 91.9 ± 3.5 | 88.0 ± 6.3 | 99.4 ± 0.2 | 41.5 ± 8.2 | 98.2 ± 0.2 | 98.6 ± 0.1 |
| R + H + C | 95.4 ± 0.4 | 96.2 ± 0.2 | 99.5 ± 0.2 | 47.2 ± 9.8 | 98.1 ± 0.3 | 98.5 ± 0.2 |

| MNIST-AddMul | DPL | | SL | | LTN | |
|---|---|---|---|---|---|---|
| | $F_1$ (**Y**) | $F_1$ (**C**) | $F_1$ (**Y**) | $F_1$ (**C**) | $F_1$ (**Y**) | $F_1$ (**C**) |
| ADD | 68.1 ± 6.7 | 0.0 ± 0.0 | 99.5 ± 0.2 | 0.0 ± 0.1 | 67.4 ± 0.1 | 0.0 ± 0.0 |
| MULT | 100.0 ± 0.0 | 37.6 ± 0.2 | 100.0 ± 0.0 | 76.1 ± 11.7 | 98.1 ± 0.5 | 78.1 ± 0.4 |
| MULTIOP | 100.0 ± 0.0 | 99.8 ± 0.1 | 100.0 ± 0.0 | 99.8 ± 0.1 | 98.3 ± 0.2 | 98.3 ± 0.2 |

*sufficient to rule out RSs, even when the data is exhaustive*. However, when forcing disentanglement, the percentage of models affected by RSs drops to *zero* for all data sets and methods, indicating that mitigation strategies that go beyond the standard learning setup – and specifically, disentanglement – can be extremely effective.

**Q2: Disentanglement is not enough under selection bias.** Next, we look at two (non-exhaustive) data sets where RS occur due to *selection bias*. Label and concept quality is measured using the $F_1$-score on the macro average measured on the test set. From here onward, all models are disentangled by construction, and despite this are affected by RSs, as shown below.

We start by evaluating `MNIST-EvenOdd`, a variant of `MNIST-Addition` (inspired by [21]) where only 16 possible pairs of digits out of 100 are given for training: 8 comprise only even digits and 8 only odd digits. As shown in [21], this setup allows to exchange the semantics of the even and odd digits while ensuring all sums are correct (because, *e.g.*, 4 + 8 = 12 = 9 + 3). As commonly done in NeSy, hyperparameters were chosen to optimize for label prediction performance on a validation set, cf. Appendix C. The impact of reconstruction (R), concept supervision (C), and Shannon entropy loss (H) on all architectures are reported in Table 3. Roughly speaking, a concept $F_1$ below 95% typically indicates a RSs, as shown by the concept confusion matrices in Appendix C.5. The main take away is that *no strategy alone can effectively mitigate RSs for any of the methods*. Additionally, for DPL and LTN, these also tend to interfere with label supervision, yielding degraded prediction performance. The SL is not affected by this, likely because it is the only method using a separate neural layer to predict the labels. Combining multiple strategies does improve concept quality on average, depending on the method. In particular, H + C and R + H + C help LTN identify good concepts, and similarly for DPL, although concept quality is slightly less stable. For the SL only concept supervision is relatively effective, but the other strategies are not, probably due to the extra flexibility granted by the top neural layer compared to DPL and LTN.

Next, we evaluate the impact of multi-task learning on an arithmetic task, denoted `MNIST-AddMul`. Here, the model observes inputs $\mathbf{x} \in \{(0, 1), (0, 2), (1, 3)\}$ and has to predict both their sum *and* their product, either separately (no MTL) or jointly (MTL). The results in Table 3 show that, as in Example 2, all methods are dramatically affected by RSs when MTL is not in place. DPL and LTN also yield sub-par $F_1$-scores on the labels for the addition task. For SL and LTN, we also observe that, despite high concept $F_1$ for the multiplication task, the digit 2 is *never* predicted correctly. This is clearly visible in the confusion matrices in Appendix C.5. However, solving addition and multiplication jointly via MTL ensures all methods acquire very high quality concepts.

**Q3: Reasoning Shortcuts are pervasive in real-world tasks.** Finally, we look at RSs occurring in `BDD-OIA` [54], an autonomous vehicle prediction task. The goal is to predict multiple possible actions $\mathbf{Y} = (\texttt{move\_forward}, \texttt{stop}, \texttt{turn\_left}, \texttt{turn\_right})$ from frames $\mathbf{x}$ of real driving scenes. Each scene is described by 21 binary concepts $\mathbf{C}$ and the knowledge K ensures the predictions are consistent with safety constraints (*e.g.*, $\texttt{stop} \Rightarrow \neg\texttt{move\_forward}$) and guarantees concepts are predicted consistently with one another (*e.g.*, $\texttt{road\_clear} \Leftrightarrow \neg\texttt{obstacle}$). Since this is a high-stakes task, we solve it with DPL, as it is the only approach out of the ones we consider that *guarantees* hard compliance with the knowledge [7]. See Appendix C for the full experimental setup.

Table 4 lists the results for DPL paired with different mitigation strategies. In order to get a sense of the model's ability to acquire good concepts, we also include two baselines: `CBM-AUC` introduced by

Table 4: **Q3**. **Left**: Means and std. deviations over 10 runs for DPL paired with different mitigations. **Right**: Confusion matrices on the training set for DPL alone (*top*) and paired with mitigation strategies (*bottom*) for {move_forward,stop,turn_left,turn_right} concept vectors.

|  | BDD-OIA | |
|---|---|---|
|  | F1-mean (Y) | F1-mean(C) |
| CBM-AUC | $70.8 \pm 0.1$ | $62.1 \pm 0.1$ |
| C-only | $64.8 \pm 0.2$ | $60.3 \pm 0.1$ |
| DPL | $71.4 \pm 0.1$ | $39.4 \pm 6.2$ |
| DPL+H | $\mathbf{72.1 \pm 0.1}$ | $48.1 \pm 0.3$ |
| DPL+C | $68.2 \pm 0.2$ | $60.5 \pm 0.1$ |
| DPL+C +H | $68.3 \pm 0.3$ | $\mathbf{61.7 \pm 0.1}$ |

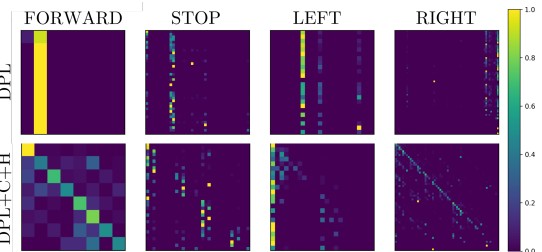

Sawada and Nakamura [55] for this task, which learns jointly the concepts together with unsupervised ones, and processes both of them with a linear layer to obtain the labels; C-only, in which the concept extractor is trained with full concept supervision and frozen, and DPL is only used to perform inference. This avoids any interference between label and concept supervision. The $F_1$-scores are computed over the test set and averaged over all 4 labels and all 21 concepts. On the right, we also report the concept confusion matrices (CMs) for the training set: each row corresponds to a ground-truth concept *vector* **g**, and each column to a predicted concept *vector* **c**.

Overall, the results show that, unless supplied with concept supervision, DPL optimizes for label accuracy ($F_1(Y)$) by leveraging low quality concepts ($F_1(C)$). This occurs even when it is paired with a Shannon entropy penalty H. This is especially evident in the CMs, which show that, for certain labels, all ground-truth concepts **g** are mapped a single **c**, not unlike what happens in our first experiment for entangled models. Conversely, the concept quality of DPL substantially improves when concept supervision is available, at the cost of a slight degradation in prediction accuracy. The CMs back up this observation as the learned concepts tend to align much closer to the diagonal. Only for the turn_left label concept supervision fails to prevent collapse. This occurs because annotations for the corresponding concepts are poor: for a large set of examples, the concepts necessary to predict turn_left $= 1$ are annotated as negatives, complicating learning. In practice, this means that all variants of DPL predict those concepts as negative, hindering concept quality.

## 7 Related Work

**Shortcuts in ML.** State-of-the-art ML predictors often achieve high performance by exploiting spurious correlations – or "shortcuts" – in the training data [56]. Well known examples include watermarks [57], background pixels [58, 59], and textual meta-data in X-ray scans [16]. Like RSs, regular shortcuts can compromise the classifier's reliability and out-of-distribution generalization and are hard to identify based on accuracy alone. Proposed solutions include dense annotations [60], out-of-domain data [61], and interaction with annotators [62]. Shortcuts are often the result of confounding resulting from, *e.g.*, selective sampling. RSs may also arise due to confounding, as is the case in MNIST-Addition (cf. Example 1), however – as discussed in Section 5, data is not the only factor underlying RSs. For instance, in XOR (cf. Example 3) RSs arise despite exhaustive data.

**Reasoning shortcuts.** The issue of RSs has so far been mostly neglected in the NeSy literature, and few remedies have been introduced but never theoretically motivated. Stammer et al. [63] have investigated shortcuts affecting NeSy architectures, but consider only *input-level* shortcuts that occur even if concepts are high-quality and fix by injecting additional knowledge in the model. In contrast, we focus on RSs that impact the quality of learned concepts. Marconato et al. [21] introduce the concept of RSs in the context of NeSy but for continual learning, and proposes a combination of concept supervision and concept-level rehearsal to address them, without delving into a theoretical justification. Li et al. [11] propose a minimax objective that ensures the concepts learned by the model satisfy K. Like the entropy regularizer [20] we addressed in Section 5, this strategy ends up spreading probability across all concepts that satisfy the knowledge, including those that have unintended semantics. As such, it does not directly address RSs.

**Neuro-symbolic integration.** While RSs affect a number of NeSy predictors, NeSy encompasses a heterogeneous family of architectures integrating learning and reasoning [1, 2]. We conjecture that

RSs do transfer to *all* NeSy approaches that do not specifically address the factors we identified, but an in-depth analysis of RSs in NeSy is beyond the scope of this paper. In a concurrent work, Wang et al. [64] derive the sample complexity of NeSy predictors when no RSs are in place. It is unclear how these bounds may change when RSs are present.

**Relation to disentanglement.** Recovering the latent variables $\mathbf{G}$ has been the center of several works in representation learning [65, 35, 66]. Among these, achieving identifiability of the latent components has been studied in non-linear independent component analysis [67, 68, 69] and recently in Causal Representation Learning [46, 70, 71, 72]. In this respect, several overlaps exist with the natural mitigation strategies that we investigated: (i) multi-task learning has been shown to increase disentanglement [73] and provably leads to identifiability [74, 75], and (ii) latent variables supervision also largely increases the amount of disentanglement [76, 77, 47]. Our work is the witness that the intersection with *disentanglement* literature is beneficial for learning the intended concepts in NeSy and, vice-versa, that knowledge-guided learning can be a new way of acquiring identifiable representations, by avoiding RSs.

## 8    Conclusion

In this work, we provide the first in-depth analysis of RS affecting NeSy predictors. Our analysis highlights four key causes of RS and suggests several mitigation strategies, which we analyze both theoretically and empirically. Our experiments indicate that RSs do naturally appear in both synthetic and real-world NeSy prediction tasks, and that the effectiveness of mitigation strategies is model and task dependent, and that a general recipe for avoiding RSs is currently missing.

We foresee that reasoning shortcuts extend beyond the current scope of NeSy predictors with known prior knowledge. This includes approaches that learn jointly both the concepts and the knowledge, like ROAP [78] and DSL [79], as well as fully neural models, like CBMs [80] and their variants [39, 81], which are also likely to be affected by RSs when concept supervision is limited. We plan to investigate further this direction in the near future.

Ultimately, this work aims at jumpstarting research on the analysis and mitigation of RSs, with the hope of leading to more trustworthy and explainable NeSy architectures.

**Broader impact.** Our work brings the subtle but critical issue of RSs to the spotlight, and highlights benefits and limitations of a variety of mitigation strategies, but it is otherwise fundamental research that has no direct societal impact.

## Acknowledgments and Disclosure of Funding

We thank the anonymous reviewers for their valuable feedback in improving the current manuscript. We acknowledge Emile von Krieken for proofreading our claims and Yoshihide Sawada for his help in deploying the models on `BDD-OIA`. We are also grateful to Pedro Zuidberg dos Martires for insightful discussions with us at the early stages of the work. We acknowledge the support of the PNRR project FAIR - Future AI Research (PE00000013), under the NRRP MUR program funded by the NextGenerationEU. The research of ST and AP was partially supported by TAILOR, a project funded by EU Horizon 2020 research and innovation programme under GA No 952215.

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

# A  Other NeSy Predictors

In this appendix, we outline the NeSy prediction approaches used in our experiments and then show that they share deterministic RSs as DPL.

The **semantic loss** (SL) [8] is a penalty term that encourages a neural network to place all probability mass on predictions that are consistent with prior knowledge K. In our setting, the SL is applied to a predictor $p_\theta(\mathbf{Y} \mid \mathbf{C})$ placed on top of a concept extractor $p_\theta(\mathbf{C} \mid \mathbf{X})$ and it can be written as:

$$\mathsf{SL}(p_\theta, (\mathbf{x}, \mathbf{y}), \mathsf{K}) := -\log \sum_{\mathbf{c}} \mathbb{1}\{(\mathbf{c}, \mathbf{y}) \models \mathsf{K}\} \, p_\theta(\mathbf{c} \mid \mathbf{x}) \tag{11}$$

Like DPL, the SL relies on knowledge compilation to efficiently implement Eq. (11). Importantly, if the distribution $p_\theta(\mathbf{c} \mid \mathbf{x})$ allocates mass *only* to concepts $\mathbf{c}$ that satisfy the knowledge given $\mathbf{y}$, these are optimal solutions, in the sense that the SL is exactly *zero*. We will make use of this fact in Appendix A.1.

During training, the SL is combined with any regular supervised loss $\ell$, for instance the cross-entropy, leading to the overall training loss:

$$\tfrac{1}{|\mathcal{D}|} \sum_{(\mathbf{x}, \mathbf{y}) \in \mathcal{D}} \ell(p_\theta, (\mathbf{x}, \mathbf{y})) + \mu \mathsf{SL}(p_\theta, (\mathbf{x}, \mathbf{y}), \mathsf{K}) \tag{12}$$

with $\mu > 0$ a hyperparameter. During inference, the SL plays no role: the predicted label is obtained by simply taking the most likely configuration through a forward pass over the network.

**Logic tensor networks** (LTNs) [6] are another state-of-the-art NeSy architecture that combines elements of reasoning- and penalty-based approaches. The core idea behind LTNs, and of all other NeSy predictors based of fuzzy logic [82], is to *relax* the prior knowledge K into a real-valued function $\mathcal{T}[\mathsf{K}]$ *quantifying* how close a prediction is to satisfying K. LTNs perform this transformation using *product real logic* [6]. In our context, this function takes the form $\mathcal{T}[\mathsf{K}] : [0, 1]^k \times [0, 1]^n \to [0, 1]$, and it takes as input the probabilities of the various concepts $\mathbf{C}$ and labels $\mathbf{Y}$ and outputs a degree of satisfaction.

Crucially, fuzzy logics are designed such that, if all probability mass is allocated to configurations $(\mathbf{c}, \mathbf{y})$ that *do* satisfy the logic, then the degree of satisfaction is exactly $1$, *i.e.*, maximal. We will leverage this fact in Appendix A.1.

During training, LTNs guide the concept extractor $p_\theta(\mathbf{C} \mid \mathbf{X})$ towards predicting concepts that satisfy the prior knowledge K by penalizing it proportionally to how far away their predictions are from satisfying K given the ground-truth label $\mathbf{y}$, that is, $1 - \mathcal{T}[\mathsf{K}](p(\mathbf{C} \mid \mathbf{x}), \mathbb{1}\{\mathbf{Y} = \mathbf{y}\})$. During inference, LTNs first predict the most likely concepts $\hat{\mathbf{c}} = \operatorname{argmax}_{\mathbf{c}} p_\theta(\mathbf{c} \mid \mathbf{x})$ using a forward pass through the network, and then predict a label $\hat{\mathbf{y}}$ that maximally satisfies the knowledge given $\hat{\mathbf{c}}$, again according to $\mathcal{T}[\mathsf{K}]$.

## A.1  Deterministic Optima are Shared

The bulk of our theoretical analysis focuses on DPL because it offers a clear probabilistic framework, however in the following we show that deterministic optima do transfer to other NeSy predictors approaches. Specifically, under **A2**, DPL, SL, and LTN all admit the same deterministic optima (det-opts).

To see this, let $\mathcal{C}_\mathbf{y}$ be the set of concepts vectors $\mathbf{c}$ from which the label $\mathbf{y}$ can be inferred, that is, $\mathcal{C}_\mathbf{y} = \{\mathbf{c} \in \mathcal{C} \, : \, p^*(\mathbf{y} \mid \mathbf{c}) > 0\}$. Now, take a deterministic RS $p_\theta(\mathbf{C} \mid \mathbf{X})$ for DPL, *i.e.*, a concept distribution that satisfies Definition 1 when the likelihood is computed as in Eq. (2). By determinism, this distribution can be equivalently written as a function mapping inputs $\mathbf{x}$ to concept vectors $\hat{\mathbf{c}}(\mathbf{x}) \in \mathcal{C}$, that is, $p_\theta(\mathbf{C} \mid \mathbf{X} = \mathbf{x}) = \mathbb{1}\{\mathbf{C} = \hat{\mathbf{c}}(\mathbf{x})\}$. At the same time, by optimality and **A2** we have that, for every $(\mathbf{x}, \mathbf{y}) \in \mathcal{D}$, the NeSy predictor $p_\theta(\mathbf{Y} \mid \mathbf{X}; \mathsf{K}) = \sum_{\mathbf{c}} u_\mathsf{K}(\mathbf{Y} \mid \mathbf{c}) \cdot p_\theta(\mathbf{c} \mid \mathbf{X})$ allocates all probability mass to the correct label $\mathbf{y}$. As a consequence, $\hat{\mathbf{c}}(\mathbf{x}) \in \mathcal{C}_\mathbf{y}$, for every $(\mathbf{x}, \mathbf{y}) \in \mathcal{D}$.

Consider a NeSy predictor that first predicts concepts using $p_\theta(\mathbf{C} \mid \mathbf{X})$ and then labels using a reasoning layer based on fuzzy logic, such that a prediction $\hat{\mathbf{y}} \in \mathcal{Y}$ is chosen so that it minimial distance from satisfaction w.r.t. K, and the distance from satisfaction is also used as a training objective. Since $\hat{\mathbf{c}} \in \mathcal{C}_\mathbf{y}$ for every $(\mathbf{x}, \mathbf{y}) \in \mathcal{D}$, by definition of T-norms, the label with minimal distance from satisfaction will necessarily be $\mathbf{y}$: all other labels cannot be inferred from the knowledge,

so they have larger distance from satisfaction. Hence, training loss will be minimal for this predictor as well, meaning that $p_\theta(\mathbf{C} \mid \mathbf{X})$ is a deterministic RS for it too.

Next, consider a NeSy penalty-based predictor where predictions are obtained by first inferring concepts using MAP over the concept extractor, that is, $\hat{\mathbf{c}} = \operatorname{argmax}_{\mathbf{c}} p_\theta(\mathbf{c} \mid \mathbf{X})$, and then performing a forward pass over a neural prediction layer $p_\theta(\mathbf{Y} \mid \mathbf{C})$. By construction, the penalty will be minimal whenever $(\mathbf{y}, \hat{\mathbf{c}}(\mathbf{x})) \models \mathsf{K}$. We already established that this is the case for all $(\mathbf{x}, \mathbf{y}) \in \mathcal{D}$, meaning that $p_\theta(\mathbf{C} \mid \mathbf{X})$ is a deterministic RS for penalty-based predictors too.

# B  Proofs

In the proofs, we suppress $\mathsf{K}$ from the notation for readability.

## B.1  Proof of [Lemma 1]: Upper Bound of the Log-Likelihood

**Proof plan.** The proof of the first point of our claim is split in three parts:

1. We show that in expectation the log-likelihood in Eq. (2) is upper bounded by a term containing the KL divergence for $p_\theta(\mathbf{Y} \mid \mathbf{G})$.

2. We prove that, under **A1**, any optimum of Eq. (2) minimizes also the KL. In this step, we make use of Information Theory [83] to connect the two.

3. We show that assuming **A1** and **A2** the optima $p_\theta(\mathbf{Y} \mid \mathbf{G})$ for the KL are given only by optima $p_\theta(\mathbf{Y} \mid \mathbf{X})$ for Eq. (2).

We proceed to prove the second point by leveraging the fact that $p_\theta(\mathbf{C} \mid \mathbf{G})$ is given by marginalizing $p_\theta(\mathbf{C} \mid \mathbf{X})$ over the generating distribution $p^*(\mathbf{X} \mid \mathbf{G}, \mathbf{S})$.

**Point (*i*).** (1) We upper bound the log-likelihood in Eq. (1) as follows:

$$\mathcal{L}(p_\theta, \mathcal{D}) = \mathbb{E}_{\mathbf{g} \sim p(\mathbf{G})} \mathbb{E}_{\mathbf{s} \sim p(\mathbf{S})} \mathbb{E}_{\mathbf{x} \sim p^*(\mathbf{X}|\mathbf{g},\mathbf{s})} \mathbb{E}_{\mathbf{y} \sim p^*(\mathbf{Y}|\mathbf{g})} \big[ \log p_\theta(\mathbf{y} \mid \mathbf{x}) \big] \tag{13}$$

$$= \mathbb{E}_{\mathbf{g} \sim p(\mathbf{G})} \mathbb{E}_{\mathbf{x} \sim p^*(\mathbf{X}|\mathbf{g})} \mathbb{E}_{\mathbf{y} \sim p^*(\mathbf{y}|\mathbf{g})} \big( \log p_\theta(\mathbf{y} \mid \mathbf{x}) \big) \tag{14}$$

$$\leq \mathbb{E}_{\mathbf{g} \sim p(\mathbf{G})} \mathbb{E}_{\mathbf{y} \sim p^*(\mathbf{Y}|\mathbf{g})} \big( \log \mathbb{E}_{\mathbf{x} \sim p^*(\mathbf{X}|\mathbf{g})} [p_\theta(\mathbf{y} \mid \mathbf{x})] \big) \tag{15}$$

$$= \mathbb{E}_{\mathbf{g} \sim p(\mathbf{G})} \mathbb{E}_{\mathbf{y} \sim p^*(\mathbf{Y}|\mathbf{g})} \big( \log p_\theta(\mathbf{y} \mid \mathbf{g}) \big) \tag{16}$$

$$= \mathbb{E}_{\mathbf{g} \sim p(\mathbf{G})} \mathbb{E}_{\mathbf{y} \sim p^*(\mathbf{Y}|\mathbf{g})} \Big( \log \frac{p_\theta(\mathbf{y} \mid \mathbf{g})}{p^*(\mathbf{y} \mid \mathbf{g})} - \log p^*(\mathbf{y} \mid \mathbf{g}) \Big) \tag{17}$$

$$= \mathbb{E}_{\mathbf{g} \sim p(\mathbf{G})} \big( - \mathsf{KL}[p^*(\mathbf{y} \mid \mathbf{g}) \| p_\theta(\mathbf{y} \mid \mathbf{g})] - \mathsf{H}[p^*(\mathbf{y} \mid \mathbf{g})] \big) \tag{18}$$

In the second line, we introduced the distribution $p^*(\mathbf{x} \mid \mathbf{g}) := \mathbb{E}_{\mathbf{s} \sim p(\mathbf{S})}[p^*(\mathbf{x} \mid \mathbf{g}, \mathbf{s})]$, in the third line we applied Jensen's inequality, and in the fifth line we added and subtracted $p^*(\mathbf{y} \mid \mathbf{g})$. This proves our claim.

(2) We proceed showing that for a deterministic $p^*(\mathbf{X} \mid \mathbf{G}, \mathbf{S})$ every optimum $p_\theta(\mathbf{Y} \mid \mathbf{X})$ for the log-likelihood leads to an optimum $p_\theta(\mathbf{Y} \mid \mathbf{G})$ for the RHS of Eq. (29). We can rewrite the joint distribution as:

$$p(\mathbf{x}, \mathbf{y}) := \mathbb{E}_{\mathbf{g} \sim p(\mathbf{G})} \mathbb{E}_{\mathbf{s} \sim p(\mathbf{S})} [p^*(\mathbf{x} \mid \mathbf{g}, \mathbf{s}) p^*(\mathbf{y} \mid \mathbf{g})] \tag{19}$$

$$= \mathbb{E}_{\mathbf{g} \sim p(\mathbf{G})} \big[ p^*(\mathbf{y} \mid \mathbf{g}) \, \mathbb{E}_{\mathbf{s} \sim p(\mathbf{S})} [p^*(\mathbf{x} \mid \mathbf{g}, \mathbf{s})] \big] \tag{20}$$

$$= \mathbb{E}_{\mathbf{g} \sim p(\mathbf{G})} \big[ p^*(\mathbf{y} \mid \mathbf{g}) p^*(\mathbf{x} \mid \mathbf{g}) \big] \tag{21}$$

$$= \mathbb{E}_{\mathbf{g} \sim p(\mathbf{G})} \big[ p^*(\mathbf{y} \mid \mathbf{g}) p^*(\mathbf{g} \mid \mathbf{x}) p(\mathbf{x}) / p(\mathbf{g}) \big] \tag{22}$$

$$= \mathbb{E}_{\mathbf{g} \sim p^*(\mathbf{G}|\mathbf{x})} [p^*(\mathbf{y} \mid \mathbf{g})] p(\mathbf{x}) \tag{23}$$

$$= p^*(\mathbf{y} \mid \mathbf{x}) p(\mathbf{x}) \tag{24}$$

where $p^*(\mathbf{y} \mid \mathbf{x}) = \mathbb{E}_{\mathbf{g} \sim p^*(\mathbf{g}|\mathbf{x})}[p^*(\mathbf{y} \mid \mathbf{g})]$ and $p^*(\mathbf{G} \mid \mathbf{x})$ is the posterior underlying the data generative process. Hence, the log-likelihood in Eq. (1) can be equivalently written as:

$$\mathbb{E}_{(\mathbf{x},\mathbf{y}) \sim p(\mathbf{X},\mathbf{Y})}[\log p_\theta(\mathbf{y} \mid \mathbf{x})] = \mathbb{E}_{\mathbf{x} \sim p(\mathbf{X})}\mathbb{E}_{\mathbf{y} \sim p^*(\mathbf{Y}|\mathbf{x})}[\log p_\theta(\mathbf{y} \mid \mathbf{x})] \tag{25}$$

$$= \mathbb{E}_{\mathbf{x} \sim p(\mathbf{X})}\mathbb{E}_{\mathbf{y} \sim p^*(\mathbf{Y}|\mathbf{x})}\Big[\log \frac{p_\theta(\mathbf{y} \mid \mathbf{x})}{p^*(\mathbf{y} \mid \mathbf{x})} - \log p^*(\mathbf{y} \mid \mathbf{x})\Big] \tag{26}$$

$$= \mathbb{E}_{\mathbf{x} \sim p(\mathbf{X})}\big(-\mathsf{KL}[p^*(\mathbf{y} \mid \mathbf{x})\|p_\theta(\mathbf{y} \mid \mathbf{x})] - \mathsf{H}[p^*(\mathbf{y} \mid \mathbf{x})]\big) \tag{27}$$

In the first line, we used $p(\mathbf{X}, \mathbf{Y}) = p^*(\mathbf{Y} \mid \mathbf{X})p(\mathbf{X})$, and then added and subtracted $\log p^*(\mathbf{y} \mid \mathbf{x})$. By comparing Eq. (18) and Eq. (27), we obtain:

$$\mathbb{E}_{\mathbf{x} \sim p(\mathbf{X})}\big(-\mathsf{KL}[p^*(\mathbf{Y} \mid \mathbf{x})\|p_\theta(\mathbf{Y} \mid \mathbf{x})] - \mathsf{H}[p^*(\mathbf{Y} \mid \mathbf{x})]\big)$$
$$\leq \mathbb{E}_{\mathbf{g} \sim p(\mathbf{G})}\big(-\mathsf{KL}[p^*(\mathbf{Y} \mid \mathbf{g})\|p_\theta(\mathbf{y} \mid \mathbf{g})] - \mathsf{H}[p^*(\mathbf{Y} \mid \mathbf{g})]\big) \tag{28}$$

Now, consider a distribution $p_\theta(\mathbf{Y} \mid \mathbf{X})$ that attains maximal likelihood. Then, $\mathsf{KL}[p^*(\mathbf{y} \mid \mathbf{x})\|p_\theta(\mathbf{y} \mid \mathbf{x})] = 0$, and we can rearrange the inequality in Eq. (28) to obtain:

$$\mathbb{E}_{\mathbf{g} \sim p(\mathbf{G})}\big(\mathsf{KL}[p^*(\mathbf{y} \mid \mathbf{g})\|p_\theta(\mathbf{y} \mid \mathbf{g})]\big) \leq \mathsf{H}[\mathbf{Y} \mid \mathbf{X}] - \mathsf{H}[\mathbf{Y} \mid \mathbf{G}] \tag{29}$$

Here, $\mathsf{H}[\mathbf{Y} \mid \mathbf{X}] = \mathbb{E}_{\mathbf{x} \sim p(\mathbf{X})}[\mathsf{H}[p^*(\mathbf{Y} \mid \mathbf{x})]$ and $\mathsf{H}[\mathbf{Y} \mid \mathbf{G}] = \mathbb{E}_{\mathbf{g} \sim p(\mathbf{G})}\mathsf{H}[p^*(\mathbf{Y} \mid \mathbf{g})]$ are conditional entropies.

We want to show that, under **A1**, the right-hand side of Eq. (29) is in fact zero. As for the other conditional entropy term, recall that [83]:

$$\mathsf{H}[\mathbf{Y} \mid \mathbf{X}] = \mathsf{H}[\mathbf{Y} \mid \mathbf{X}] - \mathsf{H}[\mathbf{Y}] + \mathsf{H}[\mathbf{Y}] = -\mathsf{I}[\mathbf{X} : \mathbf{Y}] + \mathsf{H}[\mathbf{Y}] \tag{30}$$

where $\mathsf{I}[\cdot, \cdot]$ is the mutual information. By the chain rule of the mutual information, cf. [83, Theorem 2.8.1], we have:

$$\mathsf{I}[\mathbf{Y} : \mathbf{X}, \mathbf{G}] = \mathsf{I}[\mathbf{Y} : \mathbf{X}] + \mathsf{I}[\mathbf{Y} : \mathbf{G} \mid \mathbf{X}] = \mathsf{I}[\mathbf{Y} : \mathbf{G}] + \mathsf{I}[\mathbf{X} : \mathbf{Y} \mid \mathbf{G}] \tag{31}$$

where $\mathsf{I}[\cdot, \cdot \mid \cdot]$ is the conditional mutual information. The structure of our generative process in Fig. 1 implies that $\mathsf{I}[\mathbf{X} : \mathbf{Y} \mid \mathbf{G}] = 0$, so Eq. (31) boils down to:

$$\mathsf{I}[\mathbf{X} : \mathbf{Y}] = \mathsf{I}[\mathbf{Y} : \mathbf{G}] - \mathsf{I}[\mathbf{Y} : \mathbf{G} \mid \mathbf{X}] \tag{32}$$

By **A1**, we have that:

$$\mathsf{I}[\mathbf{Y} : \mathbf{G} \mid \mathbf{X}] = \mathsf{H}[\mathbf{G} \mid \mathbf{X}] - \mathsf{H}[\mathbf{G} \mid \mathbf{X}, \mathbf{Y}] \tag{33}$$

$$= \mathsf{H}[f_{1:k}^{-1}(\mathbf{X}) \mid \mathbf{X}] - \mathsf{H}[f_{1:k}^{-1}(\mathbf{X}) \mid \mathbf{X}, \mathbf{Y}] \tag{34}$$

$$= \mathsf{H}[f_{1:k}^{-1}(\mathbf{X}) \mid \mathbf{X}] - \mathsf{H}[f_{1:k}^{-1}(\mathbf{X}) \mid \mathbf{X}] = 0 \tag{35}$$

Plugging this into Eq. (32) entails that $\mathsf{I}[\mathbf{X} : \mathbf{Y}] = \mathsf{I}[\mathbf{Y} : \mathbf{G}]$, or equivalently that $\mathsf{H}[\mathbf{Y} : \mathbf{X}] = \mathsf{H}[\mathbf{Y} : \mathbf{G}]$. This means that the right-hand side of Eq. (29) is indeed zero, which entails that the $\mathsf{KL}$ is zero and that therefore $p_\theta(\mathbf{Y} \mid \mathbf{G})$ optimizes the right-hand side of Eq. (5).

(3) We proceed showing that by assuming also **A2** whatever optimum $p_\theta(\mathbf{Y} \mid \mathbf{G})$ is identified by $p_\theta(\mathbf{Y} \mid \mathbf{X})$ that is also optimum. First, note that under **A2** $p^*(\mathbf{Y} \mid \mathbf{G})$ is deterministic, so we have $\mathsf{H}[\mathbf{Y} \mid \mathbf{G}] = 0$. With **A1** and **A2**, both $p^*(\mathbf{y} \mid \mathbf{g})$ and $p^*(\mathbf{y} \mid \mathbf{x})$ are deterministic and therefore Eq. (28) can be rewritten as:

$$\mathbb{E}_{\mathbf{x} \sim p(\mathbf{X})}[\log p_\theta(\mathbf{Y} = (\beta_\mathsf{K} \circ f_{1:k}^{-1})(\mathbf{x}) \mid \mathbf{x})] \leq \mathbb{E}_{\mathbf{g} \sim p^*(\mathbf{G})}[\log p_\theta(\mathbf{Y} = \beta_\mathsf{K}(\mathbf{g}) \mid \mathbf{g})] \tag{36}$$

In particular, for each $\mathbf{x}$ the maximum of the log-likelihood is $0$ and it is attained when the label probability is one. We make use of this observation in the next step.

Next, we show that the bound in Eq. (28) is in fact *tight*, in the sense that whenever $p_\theta(\mathbf{Y} \mid \mathbf{G})$ maximizes the left-hand side, $p_\theta(\mathbf{Y} \mid \mathbf{X})$ maximizes the right-hand side. We proceed by contradiction. Fix $\mathbf{g}$ and let $\mathcal{O}$ be the set of those $\mathbf{s}$ for which $p_\theta(\mathbf{y} \mid \mathbf{x} = f(\mathbf{g}, \mathbf{s})) < 1$ and assume that it has

non-vanishing measure. Then, the posterior distribution is also strictly less than one:

$$p_\theta(\mathbf{y} \mid \mathbf{g}) = \mathbb{E}_{\mathbf{s} \sim p(\mathbf{S})} \mathbb{E}_{\mathbf{x} \sim p^*(\mathbf{X} \mid \mathbf{g}, \mathbf{s})}[p_\theta(\mathbf{y} \mid \mathbf{x})] \tag{37}$$

$$= \int_{\mathbb{R}^q} p(\mathbf{s}) \int_{\mathbb{R}^d} p_\theta(\mathbf{y} \mid \mathbf{x}) p^*(\mathbf{x} \mid \mathbf{g}, \mathbf{s}) \mathrm{d}\mathbf{x} \mathrm{d}\mathbf{s} \tag{38}$$

$$= \int_{\mathbb{R}^s} p(\mathbf{s}) \int_{\mathbb{R}^d} \delta\{\mathbf{x} - f(\mathbf{g}, \mathbf{s})\} \mathrm{d}\mathbf{x} \mathrm{d}\mathbf{s} - \int_{\mathbb{R}^s} p(\mathbf{s}) \int_{\mathbb{R}^d} [1 - p_\theta(\mathbf{y} \mid \mathbf{x})] \delta\{\mathbf{x} - f(\mathbf{g}, \mathbf{s})\} \mathrm{d}\mathbf{x} \mathrm{d}\mathbf{s} \tag{39}$$

$$= 1 - \int_{\mathcal{O}} (1 - p_\theta(\mathbf{y} \mid \mathbf{x} = f(\mathbf{g}, \mathbf{s}))) p(\mathbf{s}) \mathrm{d}\mathbf{s} \tag{40}$$

$$< 1 \tag{41}$$

Therefore, there cannot be any optimal solutions $p_\theta(\mathbf{Y} \mid \mathbf{G})$ that are given by non-optimal probabilities $p_\theta(\mathbf{Y} \mid \mathbf{X})$. This proves the claim.

**Point (ii).** We consider now which distributions $p_\theta(\mathbf{C} \mid \mathbf{G})$ correspond to a unique distribution $p_\theta(\mathbf{C} \mid \mathbf{X})$. First, we define as $\mathcal{P}$ the set of candidate distributions $p_\varphi(\mathbf{C} \mid \mathbf{X})$, with $\varphi \in \Theta$, for which it holds:

$$\mathbb{E}_{\mathbf{s} \sim p^*(\mathbf{S})} \mathbb{E}_{\mathbf{x} \sim p^*(\mathbf{X} \mid \mathbf{g}, \mathbf{s})}[p_\varphi(\mathbf{C} \mid \mathbf{x})] = p_\theta(\mathbf{C} \mid \mathbf{g}) \tag{42}$$

For all distributions of the form $p_\theta(\mathbf{C} \mid \mathbf{G}) = \mathbb{1}\{\mathbf{C} = \mathbf{c}\}$, for $\mathbf{c} \in \mathcal{C}$, $\mathcal{P}$ restricts to a single element, i.e., $p_\varphi(\mathbf{C} \mid \mathbf{x}) = \mathbb{1}\{\mathbf{C} = \mathbf{c}\}$. We proceed by contradiction and consider a set $\mathcal{O}_\mathbf{X}$ of non-vanishing measure such that:

$$p_\varphi(\mathbf{C} \mid \mathbf{x}) \neq \mathbb{1}\{\mathbf{C} = \mathbf{c}\}, \quad \forall \mathbf{x} \in \mathcal{O}_\mathbf{X} \tag{43}$$

Let $p^*(\mathbf{X} \mid \mathbf{G}) = \mathbb{E}_{\mathbf{s} \sim p^*(\mathbf{S})} p^*(\mathbf{X} \mid \mathbf{G}, \mathbf{S})$. Then, it holds that:

$$\begin{aligned} p_\varphi(\mathbf{C} \mid \mathbf{G}) &= \int_{\mathcal{X}} p_\varphi(\mathbf{C} \mid \mathbf{x}) p^*(\mathbf{x} \mid \mathbf{G}) \mathrm{d}\mathbf{x} \\ &= \int_{\mathcal{X} \setminus \mathcal{O}_\mathbf{X}} \mathbb{1}\{\mathbf{C} = \mathbf{c}\} \, p^*(\mathbf{x} \mid \mathbf{G}) \mathrm{d}\mathbf{x} + \int_{\mathcal{O}_\mathbf{X}} p_\varphi(\mathbf{C} \mid \mathbf{x}) p^*(\mathbf{x} \mid \mathbf{G}) \mathrm{d}\mathbf{x} \\ &= (1 - \lambda) \cdot \mathbb{1}\{\mathbf{C} = \mathbf{c}\} + \lambda \cdot \tilde{p}_\varphi(\mathbf{C} \mid \mathbf{G}) \end{aligned} \tag{44}$$

where we denoted $\tilde{p}_\varphi(\mathbf{C} \mid \mathbf{G})$ the normalized probability distribution given by integrating $p_\varphi(\mathbf{C} \mid \mathbf{X})$ on $\mathcal{O}_\mathbf{X}$ solely, and $\lambda$ is the measure of $\mathcal{O}_\mathbf{X}$. Notice that the RHS of Eq. (44) is exactly $\mathbb{1}\{\mathbf{C} = \mathbf{c}\}$ *iff* $\lambda = 0$ or $\tilde{p}_\varphi(\mathbf{C} \mid \mathbf{G}) = \mathbb{1}\{\mathbf{C} = \mathbf{c}\}$, which contradicts the claim. Hence, all probabilities $p_\theta(\mathbf{C} \mid \mathbf{G}) = \mathbb{1}\{\mathbf{C} = \mathbf{c}\}$ are only given by probabilities $p_\theta(\mathbf{C} \mid \mathbf{x}) = \mathbb{1}\{\mathbf{C} = \mathbf{c}\}$, for all $\mathbf{x} \in \mathcal{X}$. This yields the claim.

## B.2 Proof of Theorem 2: Counting the Deterministic Optima

We want to count the number of deterministic optima of the log-likelihood in Eq. (1). Recall that, by Lemma 1, under **A1** and **A2** any optimum $p_\theta(\mathbf{Y} \mid \mathbf{X})$ of Eq. (2) yields an optimum $p_\theta(\mathbf{Y} \mid \mathbf{G})$ of the upper bound in Eq. (5). Following, by point (ii) of Lemma 1 we have that deterministic optima are shared between $p_\theta(\mathbf{C} \mid \mathbf{X})$ and $p_\theta(\mathbf{C} \mid \mathbf{G})$. This means that we can equivalently count the number of deterministic optima $p_\theta(\mathbf{C} \mid \mathbf{G})$ for the upper bound in Eq. (5). We proceed to do exactly this.

Let $\mathcal{A}$ be the set of all possible maps $\alpha : \mathbf{g} \mapsto \mathbf{c}$, each inducing a candidate concept distribution $p_\theta(\mathbf{C} \mid \mathbf{G}) = \mathbb{1}\{\mathbf{C} = \alpha(\mathbf{G})\}$. The only $\alpha$'s that achieve maximal likelihood are those that satisfy the knowledge for all $\mathbf{g} \in \mathsf{supp}(\mathbf{G})$ for the learning problem, that is:

$$\beta_\mathsf{K}(\mathbf{g}) = (\beta_\mathsf{K} \circ \alpha)(\mathbf{g}), \tag{45}$$

*i.e.*, that it is indeed the case that the concepts output by $\alpha(\mathbf{g})$ predict the ground-truth label $h_\mathsf{K}(\mathbf{g})$. Notice that, only one of them is correct and coincides with the identity, *i.e.*, $\alpha(\mathbf{g}) = \mathbf{g}$.

These $\alpha$ are those that satisfy the knowledge on all examples, or equivalently the *conjunction* of the knowledge applied to all examples, that is:

$$\bigwedge_{\mathbf{g} \in \mathcal{D}_\mathbf{G}} \left( (\beta_\mathsf{K} \circ \alpha)(\mathbf{g}) = h_\mathsf{K}(\mathbf{g}) \right) \tag{46}$$

This means that only a subset of $\mathcal{A}$ contains those maps consistent with the knowledge. The total number of these maps is then given by:

$$\sum_{\alpha \in \mathcal{A}} \mathbb{1}\left\{ \bigwedge_{\mathbf{g} \in \mathcal{D}_\mathbf{G}} (\beta_\mathsf{K} \circ \alpha)(\mathbf{g}) = \beta_\mathsf{K}(\mathbf{g}) \right\} \tag{47}$$

This yields the claim.

## B.3    Proof of Proposition 3: Link Between Deterministic and Non-deterministic Optima

**Point (i).** We begin by proving that, for DPL, any convex combination of optima of the likelihood Eq. (1) is itself an optimum. Fix any input $\mathbf{x}$. First, $p^*(\mathbf{y} \mid \mathbf{x})$ is the optimal value of the log-likelihood in Eq. (2), according to Eq. (29) from point (i) of Lemma 1. Now, let $p^{(1)}(\mathbf{C} \mid \mathbf{x})$ and $p^{(2)}(\mathbf{C} \mid \mathbf{x})$ be two concept distributions that both attain optimal likelihood, *i.e.*, for $i \in \{1, 2\}$ it holds that $\sum_\mathbf{c} u_\mathsf{K}(\mathbf{y} \mid \mathbf{c}) \cdot p^{(i)}(\mathbf{c} \mid \mathbf{x}) = p^*(\mathbf{y} \mid \mathbf{x})$. The likelihood term of any convex combination of the two optima is given by:

$$\sum_\mathbf{c} u_\mathsf{K}(\mathbf{y} \mid \mathbf{c})[\lambda p^{(1)}(\mathbf{c} \mid \mathbf{x}) + (1 - \lambda)p^{(2)}(\mathbf{c} \mid \mathbf{x})] \tag{48}$$

$$= \lambda \sum_\mathbf{c} u_\mathsf{K}(\mathbf{y} \mid \mathbf{c})p^{(1)}(\mathbf{c} \mid \mathbf{x}) + (1 - \lambda) \sum_\mathbf{c} u_\mathsf{K}(\mathbf{y} \mid \mathbf{c})p^{(2)}(\mathbf{c} \mid \mathbf{x}) \tag{49}$$

$$= \lambda p^*(\mathbf{y} \mid \mathbf{x}) + (1 - \lambda)p^*(\mathbf{y} \mid \mathbf{x}) \tag{50}$$

$$= p^*(\mathbf{y} \mid \mathbf{x}) \tag{51}$$

Hence, the convex combination is also an optimum of the likelihood. Note that the very same reasoning applies to the Semantic Loss (Eq. (11)), again due to linearity of the expectation over $\mathbf{C}$.

**Remark 7.** *Since SL and DPL limit $p_\theta(\mathbf{C} \mid \mathbf{X})$ to be factorized as $\prod_i p(C_i \mid \mathbf{X})$, some convex combinations that would be in principle solutions as per point (i) cannot be expressed. Essentially, this translates into an additional constraint for the solutions of SL and DPL that is, given two factorized probabilities $p^{(1)}(\mathbf{C} \mid \mathbf{x})$ and $p^{(2)}(\mathbf{C} \mid \mathbf{x})$, then:*

$$p(\mathbf{C} \mid \mathbf{x}) = \lambda p^{(1)}(\mathbf{C} \mid \mathbf{x}) + (1 - \lambda)p^{(2)}(\mathbf{C} \mid \mathbf{x}) \text{ is a solution} \iff p(\mathbf{C} \mid \mathbf{x}) \text{ is factorized } \forall x \tag{52}$$

*This applies only to models that integrate probabilistic logic by predicting the concepts independently, whereby relaxations of SL and DPL, like [10, 84], could naturally express arbitrary convex combinations of deterministic RSs.*

**Remark 8.** *The above result does not hold for LTNs, in general. We show this by constructing a counter-example for the default choice of T-conorms [6]. That is, a convex combination of deterministic optima that is not itself an optimum. Recall that LTN uses product real logic to define a degree of knowledge satisfaction and consider the prior knowledge $\mathsf{K} = (C_1 \vee C_2)$, where $C_1$ and $C_2$ are two distinct binary concepts. In LTN, the degree of satisfaction of $\mathsf{K}$ is given by the T-conorm of the logical disjunction, which is defined as $S(a, b) = a + b - ab$. Fix an input $\mathbf{x}$ and consider two deterministic distributions $p^{(1)}(\mathbf{C} \mid \mathbf{x}) = \mathbb{1}\{\mathbf{C} = (1, 0)^\top\}$ and $p^{(2)}(\mathbf{C} \mid \mathbf{x}) = \mathbb{1}\{\mathbf{C} = (0, 1)^\top\}$. It is clear that both distributions satisfy the prior knowledge and as such are optimal. Now take any convex combination $p(\mathbf{C}) = \lambda p^{(1)}(\mathbf{C}) + (1 - \lambda)p^{(2)}(\mathbf{C})$. Then, $a = \lambda p^{(1)}(C_1 = 1) + (1 - \lambda)p^{(2)}(C_1 = 1) = \lambda$ and, for similar reasons, $b = 1 - \lambda$. Then, it holds:*

$$S(p(C_1 = 1), p(C_2 = 1)) = \lambda + (1 - \lambda) - \lambda(1 - \lambda) \tag{53}$$

$$= 1 - \lambda + \lambda^2 \leq 1 \tag{54}$$

*where the equality holds iff $\lambda \in \{0, 1\}$.*

**Point (ii).** Under **A1** and **A2**, we can count all deterministic solutions $p_\theta(\mathbf{C} \mid \mathbf{G})$ via Theorem 2. Here, we show that these deterministic optima constitute a *complete* basis for all optimal solutions of Eq. (29). From (i), we have that any convex combination of the deterministic optima is also an optimum. We will show that these are the *only* optimal solutions.

Recall that the optimal solutions are all of the form $p^*(\mathbf{Y} = \beta_\mathsf{K}(\mathbf{g}) \mid \mathbf{g}) = 1$, as a consequence of **A1** and **A2** from Lemma 1, for all $\mathbf{g} \in \mathsf{supp}(\mathbf{G})$. Notice that any optimal solution $p_\theta(\mathbf{C} \mid \mathbf{g})$ must place

mass to those concepts that lead to the correct label. Formally, for every $\mathbf{g}$ and $\mathbf{y} = \beta_\mathsf{K}(\mathbf{g})$, it holds:

$$p_\theta(\mathbf{y} \mid \mathbf{g}) = \sum_\mathbf{c} u_\mathsf{K}(\mathbf{y} \mid \mathbf{c}) p_\theta(\mathbf{c} \mid \mathbf{x}) \tag{55}$$

$$\leq \sum_\mathbf{c} p_\theta(\mathbf{c} \mid \mathbf{g}) = 1 \tag{56}$$

where the equality holds *iff* $u_\mathsf{K}(\mathbf{y} \mid \mathbf{c}) = 1$ for all $p_\theta(\mathbf{c} \mid \mathbf{x}) > 0$. In other words, this means that each $\mathbf{c} \sim p_\theta(\mathbf{C} \mid \mathbf{g})$ must be an optimal solution for the logic. This proves the claim.

**Point (*iii*).** If **A2** does not hold, there can be optima solutions that are given as convex combinations of non-optimal deterministic probabilities. First, notice that according to Lemma 1, point (**i**), the optimal solutions for $p_\theta(\mathbf{Y} \mid \mathbf{X}; \mathsf{K})$ minimize the KL term and are equivalent to $p^*(\mathbf{Y} \mid \mathbf{X})$. Then, consider for a given $\mathbf{x}$ an optimal solution $p(\mathbf{C} \mid \mathbf{x}) = \lambda \mathbb{1}\{\mathbf{C} = \mathbf{c}_1\} + (1 - \lambda)\mathbb{1}\{\mathbf{C} = \mathbf{c}_2\}$ that is a convex combination of two deterministic probabilities. From the convexity of the KL it holds:

$$\mathsf{KL}[p^*(\mathbf{y} \mid \mathbf{x}) \parallel \lambda u_\mathsf{K}(\mathbf{y} \mid \mathbf{c}_1; \mathsf{K}) + (1 - \lambda)u_\mathsf{K}(\mathbf{y} \mid \mathbf{c}_2; \mathsf{K})] \tag{57}$$
$$\leq \lambda \cdot \mathsf{KL}[p^*(\mathbf{y} \mid \mathbf{x}) \parallel u_\mathsf{K}(\mathbf{y} \mid \mathbf{c}_1; \mathsf{K})] + (1 - \lambda) \cdot \mathsf{KL}[p^*(\mathbf{y} \mid \mathbf{x}) \parallel u_\mathsf{K}(\mathbf{y} \mid \mathbf{c}_2; \mathsf{K})] \tag{58}$$

where the equality holds *iff* $u_\mathsf{K}(\mathbf{y} \mid \mathbf{c}_1; \mathsf{K}) = u_\mathsf{K}(\mathbf{y} \mid \mathbf{c}_2; \mathsf{K}) = p^*(\mathbf{y} \mid \mathbf{x})$. This shows that *there can exist solutions that are convex combinations of non-optimal deterministic probabilities*. We proceed to show how the space of optimal solutions is defined. On the converse, if $u_\mathsf{K}(\mathbf{y} \mid \mathbf{c}_1; \mathsf{K})\mathbb{1}\{\mathbf{C} - \mathbf{c}_1\}$ is a solution, also $u_\mathsf{K}(\mathbf{y} \mid \mathbf{c}_2; \mathsf{K})\mathbb{1}\{\mathbf{C} - \mathbf{c}_2\}$ must be a solution:

$$\sum_\mathbf{c} u_\mathsf{K}(\mathbf{y} \mid \mathbf{c})[\lambda \mathbb{1}\{\mathbf{C} = \mathbf{c}_1\} + (1 - \lambda)\mathbb{1}\{\mathbf{C} = \mathbf{c}_2\}] = p^*(\mathbf{y} \mid \mathbf{x}) \tag{59}$$

$$\lambda u_\mathsf{K}(\mathbf{y} \mid \mathbf{c}_1) + (1 - \lambda)u_\mathsf{K}(\mathbf{y} \mid \mathbf{c}_2) = p^*(\mathbf{y} \mid \mathbf{x}) \tag{60}$$
$$\lambda p^*(\mathbf{y} \mid \mathbf{x}) + (1 - \lambda)u_\mathsf{K}(\mathbf{y} \mid \mathbf{c}_2) = p^*(\mathbf{y} \mid \mathbf{x}) \tag{61}$$
$$(1 - \lambda)u_\mathsf{K}(\mathbf{y} \mid \mathbf{c}_2) = (1 - \lambda)p(\mathbf{y} \mid \mathbf{x}) \tag{62}$$
$$u_\mathsf{K}(\mathbf{y} \mid \mathbf{c}_2) = p^*(\mathbf{y} \mid \mathbf{x}) \tag{63}$$

We now show that any optimum given by a generic $p_\varphi(\mathbf{C} \mid \mathbf{x})$, with $\varphi \in \Theta$, can be expressed as a convex combination of (1) an optimum that is a convex combination of deterministic optima and (2) an optimum that is a convex combination of deterministic, but non-optimal, probabilities:

$$p^*(\mathbf{y} \mid \mathbf{x}) = \sum_\mathbf{c} u_\mathsf{K}(\mathbf{y} \mid \mathbf{c})p(\mathbf{c} \mid \mathbf{x}) \tag{64}$$

$$= \sum_{\mathbf{c} \in \mathcal{C}_\mathbf{y}} u_\mathsf{K}(\mathbf{y} \mid \mathbf{c})p(\mathbf{c} \mid \mathbf{x}) + \sum_{\mathbf{c} \in \mathcal{S}_\mathbf{y}^c} u_\mathsf{K}(\mathbf{y} \mid \mathbf{c})p(\mathbf{c} \mid \mathbf{x}) \tag{65}$$

$$= \lambda \sum_{\mathbf{c} \in \mathcal{C}_\mathbf{y}} u_\mathsf{K}(\mathbf{y} \mid \mathbf{c})\tilde{p}(\mathbf{c} \mid \mathbf{x}) + (1 - \lambda) \sum_{\mathbf{c} \in \mathcal{S}_\mathbf{y}^c} u_\mathsf{K}(\mathbf{y} \mid \mathbf{c})\bar{p}(\mathbf{c} \mid \mathbf{x}) \tag{66}$$

where $\mathcal{C}_\mathbf{y} = \{\mathbf{c} \in \mathcal{C} : u_\mathsf{K} = p^*(\mathbf{y} \mid \mathbf{x})\}$ and $\mathcal{S}_\mathbf{y}^c = \mathcal{C}/\mathcal{C}_\mathbf{y}$ are two disjoint sets. In the second line, we rewrote the summation on the two terms considering the two sets, whereas in the third line we introduced: $\lambda = \sum_{\mathbf{c} \in \mathcal{C}_\mathbf{y}} p(\mathbf{c} \mid \mathbf{x})$ and $\tilde{p}(\mathbf{c} \mid \mathbf{x}) = p(\mathbf{c} \mid \mathbf{x})/\lambda$, $1 - \lambda = \sum_{\mathbf{c} \in \mathcal{S}_\mathbf{y}} p(\mathbf{c} \mid \mathbf{x})$ and $\bar{p}(\mathbf{c} \mid \mathbf{x}) = p(\mathbf{c} \mid \mathbf{x})/(1 - \lambda)$. Since each $\tilde{p}(\mathbf{c} \mid \mathbf{x})$ lead to an optimum by construction, it must be that also $\bar{p}(\mathbf{c} \mid \mathbf{x})$ is an optimum, by the previous point. In general, there can be many $\bar{p}(\mathbf{c} \mid \mathbf{x})$ leading to optima, even when $\tilde{p}(\mathbf{c} \mid \mathbf{x})$ reduces to only the ground-truth element, *i.e.*, $\tilde{p}(\mathbf{c} \mid \mathbf{x}) = \mathbb{1}\{\mathbf{C} - f_{1:k}^{-1}(\mathbf{x})\}$. Notice that this must hold for all $\mathbf{x} \sim p^*(\mathbf{X})$. This proves the claim.

### B.4   Proof of Proposition 4: Multi-task Learning

When considering multiple task $T$, suppose that, for each $\mathbf{x} \in \bigcap_t \mathcal{D}_t$, we get $T$ different labels $\mathbf{y}^{(1)}, \dots, \mathbf{y}^{(T)}$, each given in accordance to knowledge $\mathsf{K}^{(1)}, \dots, \mathsf{K}^{(T)}$, respectively. Similarly to Eq. (2), we consider the learning objective for all tasks with the joint log-likelihood term:

$$\log \prod_{t=1}^T p_\theta(\mathbf{y}^{(t)} \mid \mathbf{x}; \mathsf{K}^{(t)}) = \sum_{t=1}^T \log p_\theta(\mathbf{y}^{(t)} \mid \mathbf{x}; \mathsf{K}^{(t)}) \tag{67}$$

Under **A1** and **A2**, Lemma 1 point (*i*) holds for each task $t$. In the following, we denote with $\beta_{\mathsf{K}}^{(t)}$ the underlying maps for each $\mathsf{K}^{(t)}$, which by **A2** are deterministic. Hence, the learning objective becomes:

$$\mathcal{L} = \sum_{t=1}^{T} \mathbb{E}_{\mathbf{x} \sim p(\mathbf{X})} [\log p_\theta(\mathbf{Y}^{(t)} = (\beta_{\mathsf{K}}^{(t)} \circ f_{1:k}^{-1})(\mathbf{x}) \mid \mathbf{x}; \mathsf{K}^{(t)})] \tag{68}$$

and, similarly to point (*i*) of Lemma 1, we get the following upper-bound:

$$\mathcal{L} \leq \sum_{t=1}^{T} \mathbb{E}_{\mathbf{g} \sim p(\mathbf{G})} [\log p_\theta(\mathbf{Y}^{(t)} = \beta_{\mathsf{K}}^{(t)}(\mathbf{g}) \mid \mathbf{g}; \mathsf{K}^{(t)})] \tag{69}$$

Notice that the optimal values for Eq. (69) are given by those distributions that are one for each term in the summation. Then, following from Theorem 2, we have that the deterministic maps $\alpha$'s that optimize Eq. (69) must be consistent with each task $t$ and for each $\mathbf{g} \in \mathsf{supp}(\mathbf{G})$ satisfy:

$$\bigwedge_{t=1}^{T} \left( (\beta_{\mathsf{K}}^{(t)} \circ \alpha)(\mathbf{g}) = \beta_{\mathsf{K}}^{(t)}(\mathbf{g}) \right) \tag{70}$$

which, equivalently to Theorem 2, points to the condition that the maps $\alpha$ must be consistent with solving all tasks $t$. This exactly amounts to solving the conjunction of all knowledge $\mathsf{K} = \bigwedge_t \mathsf{K}^{(t)}$. This yields the claim.

## B.5    Proof of Proposition 5: Concept Supervision

Let $p^*(\mathbf{G} \mid \mathcal{S})$ be the distribution of ground-truth concepts restricted to a subset $\mathcal{S} \subseteq \mathsf{supp}(\mathbf{G})$:

$$p^*(\mathbf{G} \mid \mathcal{S}) = \frac{1}{\mathcal{Z}} p^*(\mathbf{g}) \mathbb{1}\{\mathbf{g} \in \mathcal{S}\} \quad \text{with } \mathcal{Z} = \sum_{\mathbf{g}} p^*(\mathbf{g}) \mathbb{1}\{\mathbf{g} \in \mathcal{S}\} \tag{71}$$

and $p^*(\mathbf{X} \mid \mathcal{S}) = \mathbb{E}_{(\mathbf{g},\mathbf{s}) \sim p^*(\mathbf{G}|\mathcal{S}) p^*(\mathbf{S})} p^*(\mathbf{X} \mid \mathbf{g}, \mathbf{s})$ be the corresponding restricted input distribution.

Under **A1**, the expectation of the log-likelihood term for concept supervision in Section 5 is:

$$\mathbb{E}_{\mathbf{x} \sim p^*(\mathbf{X}|\mathcal{S})} \log p_\theta(\mathbf{C}_I = f_I^{-1}(\mathbf{x}) \mid \mathbf{x}) = \mathbb{E}_{\mathbf{g} \sim p^*(\mathbf{G}|\mathcal{S})} \mathbb{E}_{\mathbf{s} \sim p^*(\mathbf{S})} \log p_\theta(\mathbf{C}_I = \mathbf{g}_I \mid f(\mathbf{g}, \mathbf{s}))$$
$$\leq \mathbb{E}_{\mathbf{g} \sim p^*(\mathbf{G}|\mathcal{S})} \log \mathbb{E}_{\mathbf{s} \sim p^*(\mathbf{S})} [p_\theta(\mathbf{C}_I = \mathbf{g}_I \mid f(\mathbf{g}, \mathbf{s}))]$$
$$= \mathbb{E}_{\mathbf{g} \sim p^*(\mathbf{G}|\mathcal{S})} \log p_\theta(\mathbf{C}_I = \mathbf{g}_I \mid \mathbf{g}) \tag{72}$$

where $p^*(\mathbf{X} \mid \mathcal{S}) = \mathbb{E}_{\mathbf{g} \sim p^*(\mathbf{G}|\mathcal{S})} \mathbb{E}_{\mathbf{s} \sim p^*(\mathbf{S})} p_\theta(\mathbf{X} \mid \mathbf{g}, \mathbf{s})$. In the first line we made use of **A1** to write $\mathbf{x} = f(\mathbf{g}, \mathbf{s})$, in the second line we used Jensen's inequality, and then we introduced the marginal distribution $p_\theta(\mathbf{C} \mid \mathbf{G}) = \mathbb{E}_{\mathbf{s} \sim p^*(\mathbf{S})} [p_\theta(\mathbf{C} \mid f(\mathbf{G}, \mathbf{s}))]$. Recall that $I$ denotes the subset of the supervised ground-truth concepts and that the log-likelihood is evaluated only w.r.t. those concepts.

Now notice that, given **A1**, the maximum for the LHS of Eq. (72) is zero, *i.e.*, coincides with maximum log-likelihood. Consistently the RHS of Eq. (72) is zero only when $p_\theta(\mathbf{C}_I \mid \mathbf{g})$ places all mass on $\mathbf{g}_I$. We proceed considering those maps $\alpha : \mathbf{g} \mapsto \mathbf{c}$ which lead to deterministic, optimal solutions for the RHS. For these, it must hold that for each $\mathbf{g} \in \mathcal{S}$:

$$\bigwedge_{i \in I} \alpha_i(\mathbf{g}) = g_i \tag{73}$$

When taken all together, we can estimate how many $\alpha \in \mathcal{A}$ (cf. Appendix B.2) satisfy the above condition for all $\mathbf{g}$:

$$\sum_{\alpha \in \mathcal{A}} \mathbb{1}\left\{ \bigwedge_{\mathbf{g} \in \mathcal{S}} \bigwedge_{i \in I} \alpha_i(\mathbf{g}) = g_i \right\} \tag{74}$$

This yields the claim.

## B.6    Proof of Proposition 6: Reconstruction

Under **A1**, the reconstruction penalty can be written as:

$$\mathbb{E}_{\mathbf{x} \sim p^*(\mathbf{X})} [\mathcal{R}(\mathbf{x})] = -\mathbb{E}_{(\mathbf{g},\mathbf{s}) \sim p^*(\mathbf{G}) p^*(\mathbf{S})} \mathbb{E}_{\mathbf{x} \sim p^*(\mathbf{X}|\mathbf{g},\mathbf{s})} [\mathbb{E}_{(\mathbf{c},\mathbf{z}) \sim p_\theta(\mathbf{C},\mathbf{Z}|\mathbf{x})} \log p_\psi(\mathbf{x} \mid \mathbf{c}, \mathbf{z})] \tag{75}$$
$$= -\mathbb{E}_{(\mathbf{g},\mathbf{s}) \sim p^*(\mathbf{G}) p^*(\mathbf{S})} [\mathbb{E}_{(\mathbf{c},\mathbf{z}) \sim p_\theta(\mathbf{C},\mathbf{Z}|f(\mathbf{g},\mathbf{s}))} [\log p_\psi(f(\mathbf{g}, \mathbf{s}) \mid \mathbf{c}, \mathbf{z})]] \tag{76}$$

From **A3**, we have that the both encoder and decoder distributions are factorized:

$$p_\theta(\mathbf{c}, \mathbf{z} \mid f(\mathbf{g}, \mathbf{s})) = p_\theta(\mathbf{c} \mid \mathbf{g})p_\theta(\mathbf{z} \mid \mathbf{s}) \tag{77}$$

$$p_\psi(f(\mathbf{g}, \mathbf{s}) \mid \mathbf{c}, \mathbf{z}) = p_\psi(\mathbf{g} \mid \mathbf{c})p_\psi(\mathbf{s} \mid \mathbf{z}) \tag{78}$$

This yields:

$$\begin{aligned}
\mathbb{E}_{\mathbf{x} \sim p(\mathbf{X})}[\mathcal{R}(\mathbf{x})] &= -\mathbb{E}_{(\mathbf{g},\mathbf{s}) \sim p^*(\mathbf{G})p^*(\mathbf{S})}\left[\mathbb{E}_{\mathbf{c} \sim p_\theta(\mathbf{C}|\mathbf{g})}[\log p_\psi(\mathbf{g}|\mathbf{c})] + \mathbb{E}_{(\mathbf{z}) \sim p_\theta(\mathbf{Z}|\mathbf{s})}[\log p_\psi(\mathbf{s}|\mathbf{z})]\right] \\
&= -\mathbb{E}_{\mathbf{g} \sim p^*(\mathbf{G})}\mathbb{E}_{\mathbf{c} \sim p_\theta(\mathbf{C}|\mathbf{g})}\log p_\psi(\mathbf{g}|\mathbf{c}) - \mathbb{E}_{\mathbf{s} \sim p^*(\mathbf{S})}\mathbb{E}_{\mathbf{z} \sim p_\theta(\mathbf{Z}|\mathbf{s})}\log p_\psi(\mathbf{s}|\mathbf{z}) \\
&\geq -\mathbb{E}_{\mathbf{g} \sim p^*(\mathbf{G})}\mathbb{E}_{\mathbf{c} \sim p_\theta(\mathbf{C}|\mathbf{g})}\log p_\psi(\mathbf{g}|\mathbf{c}) \tag{79}
\end{aligned}$$

where in the first row we separated the two logarithms and removed the expectations on $p_\theta(\mathbf{Z} \mid \mathbf{s})$ and $p_\theta(\mathbf{C} \mid \mathbf{g})$ for the terms $p_\psi(\mathbf{g} \mid \mathbf{c})$ and $p_\psi(\mathbf{s} \mid \mathbf{z})$, respectively. In the third line, we discarded the term with the match of the reconstruction of the style, giving the lower bound.

When $\mathcal{R}$ goes to zero, the lower bound also goes to zero and the last term of Eq. (79) is minimized. This happens whenever, for each $\mathbf{c} \sim p_\theta(\mathbf{c} \mid \mathbf{g})$, it holds that $p_\psi(\mathbf{g} \mid \mathbf{c}) = 1$. This condition also implies that if for two different $\mathbf{g}'$ and $\mathbf{g}''$ there exist at least one $\mathbf{c}$ such that $p_\theta(\mathbf{c} \mid \mathbf{g}') \cdot p_\theta(\mathbf{c} \mid \mathbf{g}'') > 0$, then $p_\psi(\mathbf{g}' \mid \mathbf{c})$ and $p_\psi(\mathbf{g}'' \mid \mathbf{c})$ cannot both be optimal.

We restrict now to a deterministic map $\alpha : \mathbf{g} \mapsto \mathbf{c}$ for $p_\theta(\mathbf{C} \mid \mathbf{G})$ and describe the condition when optimal decoders are attained, *i.e.*, $p_\psi(\mathbf{g} \mid \mathbf{c}) = 1$ for some $\mathbf{c}$. By the previous argument, an optimal map $\alpha$ that leads to optimal decoders must always map ground-truth concepts $\mathbf{g}$ to different concepts $\mathbf{c}$, that is:

$$\alpha(\mathbf{g}') \neq \alpha(\mathbf{g}'') \qquad \forall \mathbf{g}' \neq \mathbf{g}'' \tag{80}$$

In particular, this condition must hold for all different $\mathbf{g} \in \mathsf{supp}(\mathbf{G})$. The number of all solutions is then given by:

$$\sum_{\alpha \in \mathcal{A}} \mathbb{1}\left\{\bigwedge_{\mathbf{g} \in \mathsf{supp}_{\mathbf{G}}} \bigwedge_{\mathbf{g}' \in \mathsf{supp}_{\mathbf{G}}:\mathbf{g}' \neq \mathbf{g}} \alpha(\mathbf{g}) \neq \alpha(\mathbf{g}')\right\} \tag{81}$$

as claimed.

## C  Experimental Details and Further Results

We report here all further details concerning the experiments in Section 6.

### C.1  Implementation

The code of the experiments builds on top of `nesy-cl` [21] and `CBM-AUC` [55]. All experiments are implemented with Python 3.8.16 and Pytorch [85] and run over one A100 GPU. The code for the experiments is available at github.com/reasoning-shortcuts.

The implementation of DPL is taken from [21] and follows exactly Eq. (1), where to each world a label is assigned accordingly to the prior knowledge K. We implemented SL following the original paper [8], with the only difference that the prediction of the labels $\mathbf{Y}$ is done on top of the logits of $p_\theta(\mathbf{C} \mid \mathbf{x})$. This relaxes the conditional independence between labels and concepts while being more in line with the generative process we assumed. The implementation of LTN is adapted from `LTN-pytorch` [86] and is based on the satisfaction loss introduced in Appendix A.

### C.2  Data sets & Count of the Reasoning Shortcuts

Here, we illustrate how to count explicitly the number of deterministic RSs using the equations in Table 1, restricting ourselves to the case where no disentanglement is in place, for simplicity. Similarly to Section 4, we assume **A1** and **A2** and that $\mathsf{supp}(\mathbf{G}) = \mathcal{C} = \mathcal{G}$, and count the total number of deterministic optima (or *det-opt*s for short) under different mitigation strategies. In the following, $\mathcal{C}_\mathbf{y} \subseteq \mathcal{C} = \mathcal{G}$ refers to the set of $\mathbf{c} \in \mathcal{C}$ or $\mathbf{g} \in \mathcal{G}$ that are mapped by K to label $\mathbf{y} \in \mathcal{Y}$. Note that $\sum_\mathbf{y} |\mathcal{C}_\mathbf{y}| = |\mathcal{G}|$.

**Explicit count for the likelihood.** When the concept extractor $p_\theta(\mathbf{C} \mid \mathbf{X})$ is sufficiently expressive, the deterministic mappings $\alpha$ are essentially arbitrary. Specifically, the set $\mathcal{A}$ of these $\alpha$'s includes all functions from $\mathcal{G}$ to $\mathcal{C}$. These functions can be explicitly enumerated by counting how many ways there are to map each input vector $\mathbf{g}$ to an arbitrary vector $\mathbf{c}$.

Here, we are interested in counting the number of $\alpha$'s that attain optimal likelihood. Each such $\alpha$ has to ensure that each $\mathbf{g} \in \mathcal{G}$ is mapped to a $\mathbf{c} \in \mathcal{C}_{\beta_{\mathsf{K}}(\mathbf{g})}$ that yields the correct label, or in short, $\forall \mathbf{g} \, . \, \alpha(\mathbf{g}) \in \mathcal{C}_{\beta_{\mathsf{K}}(\mathbf{g})}$. This condition is satisfied if and only if, for every $\mathbf{y}$ in the data set, every $\mathbf{g} \in \mathcal{C}_{\mathbf{y}}$ is mapped to a $\mathbf{c}$ that is also in $\mathcal{C}_{\mathbf{y}}$, and there are exactly $|\mathcal{C}_{\mathbf{y}}|^{|\mathcal{C}_{\mathbf{y}}|}$ ways to do this. This immediately shows that the overall number of *det-opt*s $\alpha$ is:

$$\sum_{\alpha \in \mathcal{A}} \mathbb{1}\left\{\bigwedge_{\mathbf{g} \in \mathcal{G}} (\beta \circ \alpha)(\mathbf{g}) = \alpha(\mathbf{g})\right\} = \prod_{\mathbf{y} \in \mathcal{Y}} |\mathcal{C}_{\mathbf{y}}|^{|\mathcal{C}_{\mathbf{y}}|} \tag{82}$$

This associates an explicit number to Theorem 2.

**Explicit count for the reconstruction.** The effect of adding a reconstruction penalty is that now, in order to achieve optimal loss, $\alpha$ has to map different $\mathbf{g}$'s to distinct concepts $\mathbf{c}$'s. Unless this is the case, it becomes impossible to reconstruct the ground-truth concepts from the learned ones, and therefore the input $\mathbf{x}$ generated from them.

The resulting computation is the same as above, except that now we have to associate the different $\mathbf{g}$'s to different $\mathbf{c}$'s *without replacement*. This yields the following count:

$$\sum_{\alpha \in \mathcal{A}} \mathbb{1}\left\{\bigwedge_{\mathbf{g} \in \mathcal{G}} (\beta \circ \alpha)(\mathbf{g}) = \alpha(\mathbf{g})\right\} \cdot \mathbb{1}\left\{\bigwedge_{\mathbf{g}, \mathbf{g}' \in \mathcal{G} : \mathbf{g} \neq \mathbf{g}'} \alpha(\mathbf{g}) \neq \alpha(\mathbf{g}')\right\} = \prod_{\mathbf{y} \in \mathcal{Y}} |\mathcal{C}_{\mathbf{y}}|! \tag{83}$$

**Explicit count for concept supervision.** For the combination of logic and concept supervision, we suppose that for each equivalence class $\mathcal{C}_{\mathbf{y}}$, there are $\nu_{\mathbf{y}}$ ground-truth concepts provided with supervision and that all concept dimensions $C_i$ receive this supervision. Let $\mathcal{S} \subset \mathcal{G}$ be the set of supervised concepts, such that $|\mathcal{S}| = \sum_{\mathbf{y} \in \mathcal{Y}} \nu_{\mathbf{y}}$. From above, this means that we are specifying a total of $|\mathcal{S}|$ ground-truth concepts. From Table 1, we have that the number of *det-opt*s amounts to:

$$\sum_{\alpha \in \mathcal{A}} \mathbb{1}\left\{\bigwedge_{\mathbf{g} \in \mathcal{G}} (\beta \circ \alpha)(\mathbf{g}) = \alpha(\mathbf{g})\right\} \cdot \mathbb{1}\left\{\bigwedge_{\mathbf{g} \in \mathcal{S}} \bigwedge_{i=1}^{k} \alpha_i(\mathbf{g}) = g_i\right\} = \prod_{\mathbf{y} \in \mathcal{Y}} |\mathcal{C}_{\mathbf{y}}|^{|\mathcal{C}_{\mathbf{y}} - \nu_{\mathbf{y}}|} \tag{84}$$

Finally, combining together the terms of prediction, reconstruction, and concept supervision, we obtain:

$$\sum_{\alpha \in \mathcal{A}} \mathbb{1}\left\{\bigwedge_{\mathbf{g} \in \mathcal{G}} (\beta \circ \alpha)(\mathbf{g}) = \alpha(\mathbf{g})\right\} \times \tag{85}$$

$$\times \mathbb{1}\left\{\bigwedge_{\mathbf{g}, \mathbf{g}' \in \mathcal{G} : \mathbf{g} \neq \mathbf{g}'} \alpha(\mathbf{g}) \neq \alpha(\mathbf{g}')\right\} \times \tag{86}$$

$$\times \mathbb{1}\left\{\bigwedge_{\mathbf{g} \in \mathcal{S}} \bigwedge_{i=1}^{k} \alpha_i(\mathbf{g}) = g_i\right\} = \prod_{\mathbf{y} \in \mathcal{Y}} |\mathcal{C}_{\mathbf{y}} - \nu_{\mathbf{y}}|! \tag{87}$$

**The disentangled case.** If the network is *disentangled*, the enumeration procedure becomes substantially more complicated and cannot be written compactly in closed form. While the number of deterministic optima $\alpha$ can still be computed exactly using model counting [29, 30], doing so is not necessary for the scope of our paper and therefore left to future work.

### C.2.1 Dataset: XOR

This dataset, introduced in Example 3, is a toy data set containing 3 bits $\mathbf{g} = (g_1, g_2, g_3)$, for a total of 8 possible combinations. The task consists in predicting the XOR operation among them, namely $y = (g_1 \oplus g_2 \oplus g_3)$. The dataset is exhaustive and has no validation and test set. The model performances are evaluated on the training set.

**Reasoning shortcut.** For this dataset, we have $|\mathcal{C}_0| = |\mathcal{C}_1| = 4$. RSs arise depending on the structure of the underlying network. When the ground-truth concepts are processed all together without any mitigation strategy, we obtain that the number of *det-opt*s amounts to:

$$\prod_{y \in \mathcal{Y}} |\mathcal{C}_y|^{|\mathcal{C}_y|} = 4^4 \cdot 4^4 \tag{88}$$

The confusion matrices for all methods are reported in Appendix C.5. On the other hand, we show that in the *disentangled* case, only two combinations suffice to identify the correct solution, *e.g.*,

$$\begin{cases} \alpha(0) \oplus \alpha(0) \oplus \alpha(0) = 0 \\ \alpha(0) \oplus \alpha(0) \oplus \alpha(1) = 1 \end{cases} \tag{89}$$

and here the only viable solution for the two is $\alpha(0) = 0$ and $\alpha(1) = 1$. This condition is met in all our experiments in Table 2.

### C.2.2 Dataset: `MNIST-Addition`

We consider the version introduced in [7], which consists of couples of digits, each ranging from 0 to 9, and the target consists in predicting the correct sum, *i.e.*, $y = g_1 + g_2$. This data set contains all possible combinations, for a total of 100. The training set contains 42k data, the validation set 12k, and the test set 6k.

**Reasoning shortcuts.** RSs arise only as a result of the joint prediction of both digits. Notice that the number of elements $|\mathcal{C}_y|$ for each sum **y** can be evaluated as:

$$|\mathcal{C}_y| = \begin{cases} y + 1, & \text{if } y \le 9 \\ (18 - y) + 1, & \text{otherwise.} \end{cases} \tag{90}$$

Therefore, the total number of *det-opt*s amounts to:

$$\prod_{y \in \mathcal{Y}} |\mathcal{C}_y|^{|\mathcal{C}_y|} = \prod_{y=1}^{9} y^{2y} \cdot 10^{10} \tag{91}$$

When providing *disentanglement*, the number of RSs reduce to 0, as it sufficient to have the sums:

$$\alpha(c) + \alpha(c) = 2 \cdot c \tag{92}$$

to uniquely identify the value of the digit $c$.

### C.2.3 Dataset: `MNIST-EvenOdd`

This data set, proposed by [21], is a biased version of `MNIST-Addition`, where only some combinations of the digits appear. We consider here a more challenging scenario w.r.t. the proposed version, consisting of the sums:

$$\begin{cases} \boxed{0} + \boxed{6} = 6 \\ \boxed{2} + \boxed{8} = 10 \\ \boxed{4} + \boxed{6} = 10 \\ \boxed{4} + \boxed{8} = 12 \end{cases} \land \begin{cases} \boxed{1} + \boxed{5} = 6 \\ \boxed{3} + \boxed{7} = 10 \\ \boxed{1} + \boxed{9} = 10 \\ \boxed{3} + \boxed{9} = 12 \end{cases} \tag{93}$$

Overall, the training set contains 6720 data, the validation set 1920, and the test set 960.

**Reasoning shortcuts.** We describe the RSs that arise even when the architecture incorporates *disentanglement*. We evaluate the possible RSs empirically noticing that the system of observed sums can be written as a linear system, as done by Marconato et al. [21]:

$$\begin{cases} \alpha(0) + \alpha(6) = 6 \\ \alpha(2) + \alpha(8) = 10 \\ \alpha(4) + \alpha(6) = 10 \\ \alpha(4) + \alpha(8) = 12 \end{cases} \land \begin{cases} \alpha(1) + \alpha(5) = 6 \\ \alpha(3) + \alpha(7) = 10 \\ \alpha(1) + \alpha(9) = 10 \\ \alpha(3) + \alpha(9) = 12 \end{cases} \tag{94}$$

Now, notice that we can find independent reasoning shortcuts for each of the two sides since they do not share any digits. For the LHS, we consider the sum $\alpha(2) + \alpha(8) = 10$ and notice that we can find at most 10 different attributions for having a correct sum. Notice that, some of them are not allowed as $\alpha(8) = 0, 1$ leads to inconsistent values for the fourth sum, and $\alpha(8) = 3$ leads to an inconsistent sum for the first equation. So in total, for the LHS, we obtain 7 possible solutions and, by symmetry, the same number also for the LHS. In total, the number of *det-opt*s is equal to $7 \cdot 7$.

Experimentally, we consider limited concept supervision on $I = \{4, 9\}$ which should be sufficient, in principle, to disambiguate between even and odds digits. This happens because specifying $\alpha(4) = 4$ and $\alpha(9) = 9$ admits only the ground-truth concepts as the optimal solution.

### C.2.4 Dataset: `MNIST-AddMul`

In this dataset, we consider fewer combinations of digits, explicitly:

$$\begin{cases} \boxed{0} + \boxed{1} = 1 \\ \boxed{0} + \boxed{2} = 2 \\ \boxed{1} + \boxed{3} = 4 \end{cases} \tag{95}$$

and similarly for multiplication. This data set contains $1680$ training examples, $480$ for the validation, and $240$ for the test set.

**Reasoning shortcuts.** For the case of the addition task, we can have 2 possible solutions, which are:

- $\alpha(0) = 0$, $\alpha(1) = 1$, $\alpha(2) = 2$, and $\alpha(3) = 3$;
- $\alpha(0) = 1$, $\alpha(1) = 0$, $\alpha(2) = 1$, and $\alpha(3) = 4$.

For multiplication, we have that since $\alpha(1) \cdot \alpha(3) = 3$ it can be either $\alpha(1) = 1$ or $\alpha(1) = 3$. In both cases, it holds that $\alpha(0) = 0$ and $\alpha(2)$ can be arbitrary. Hence, there are in total $2 \cdot 4$ possible *det-opt*s. In MTL, the reasoning shortcut for the addition does not hold since it leads to a sub-optimal solution for multiplication.

### C.2.5 Dataset: `BDD-OIA`

This data set contains frames of driving scene videos for autonomous predictions [54]. Each frame is annotated with 4 binary labels, indicating the possible actions, $\mathbf{Y} = (\texttt{move\_forward}, \texttt{stop}, \texttt{turn\_left}, \texttt{turn\_right})$. Each scene is also annotated with 21 binary concepts $\mathbf{C}$, underlying the *reasons* for the possible actions, see Table 5. The training set contains 16k frames, with full label and concept supervision; the validation and the test set contain 2k and 4.5k annotated data, respectively.

For designing the prior knowledge w.r.t. to the concepts in Table 5, we make use of the following rules for `move_forward`/`stop` *predictions*:

$$\begin{cases} \texttt{red\_light} \implies \neg\texttt{green\_light} \\ \texttt{obstacle} = \texttt{car} \lor \texttt{person} \lor \texttt{rider} \lor \texttt{other\_obstacle} \\ \texttt{road\_clear} \iff \neg\texttt{obstacle} \\ \texttt{green\_light} \lor \texttt{follow} \lor \texttt{clear} \implies \texttt{move\_forward} \\ \texttt{red\_light} \lor \texttt{stop\_sign} \lor \texttt{obstacle} \implies \texttt{stop} \\ \texttt{stop} \implies \neg\texttt{move\_forward} \end{cases} \tag{96}$$

For `turn_left`, and similarly for `turn_right`, we use:

$$\begin{cases} \texttt{can\_turn} = \texttt{left\_lane} \lor \texttt{left\_green\_light} \lor \texttt{left\_follow} \\ \texttt{cannot\_turn} = \texttt{no\_left\_lane} \lor \texttt{left\_obstacle} \lor \texttt{left\_solid\_line} \\ \texttt{can\_turn} \land \neg\texttt{cannot\_turn} \implies \texttt{turn\_left} \end{cases} \tag{97}$$

Notice that, since the concepts are predicted together, as explained in Appendix C.4, we can count the number of RSs as follows:

- For `move_forward` and `stop`, the labels are predicted with the constraints such that $(\texttt{move\_forward}, \texttt{stop}) = (1, 1)$ has no support. Hence, we consider only the predictions $(0, 0), (0, 1), (1, 0)$. Next, we identify:
  - $|\mathcal{C}_{0,0}| = 1$, since it corresponds just to the case where no concepts are predicted.
  - $|\mathcal{C}_{(1,0)}| = 2^3 - 1$, which is the number of different concepts attribution for forward yielding a positive label;
  - $|\mathcal{C}_{(0,1)}| = 280$ are the combination of the remaining concepts that yield the `stop` action, in agreement with the constraints. These were counted explicitly from the logic implementation.

  Overall, the number of *det-opt*s amount to $1 \cdot 7^7 \cdot 280^{280}$.

- For `turn_left` and `turn_right`, we count the cardinality of positive and negative predictions of the two classes $|\mathcal{C}_0|$ and $|\mathcal{C}_1|$:

- $|\mathcal{C}_1| = 2^3 - 1$, that are the only concepts attributions for the positive label;
- $|\mathcal{C}_0| = 2^6 - |\mathcal{C}_1|$, are all the remaining concept combinations.

For left and right, separately, we obtain that possible optimal solutions amount to $7^7 \cdot 57^{57}$.

Altogether, the count of *det-opt*s for BDD-OIA goes as follows:

$$\prod_{\mathbf{y} \in \mathcal{Y}} |\mathcal{C}_\mathbf{y}|^{|\mathcal{C}_\mathbf{y}|} = 1^1 \cdot 7^7 \cdot 280^{280} \cdot 7^7 \cdot 57^{57} \cdot 7^7 \cdot 57^{57} \tag{98}$$

Table 5: Concepts annotated in BDD-OIA. Table taken from [54]

| Action Category | Concepts | Count |
|---|---|---|
| move_forward | green_light | 7805 |
| | follow | 3489 |
| | road_clear | 4838 |
| stop | red_light | 5381 |
| | traffic_sign | 1539 |
| | car | 233 |
| | person | 163 |
| | rider | 5255 |
| | other_obstacle | 455 |
| turn_left | left_lane | 154 |
| | left_green_light | 885 |
| | left_follow | 365 |
| | no_left_lane | 150 |
| | left_obstacle | 666 |
| | letf_solid_line | 316 |
| turn_right | right_lane | 6081 |
| | right_green_light | 4022 |
| | right_follow | 2161 |
| | no_right_lane | 4503 |
| | right_obstacle | 4514 |
| | right_solid_line | 3660 |

### C.3   Optimizer and Hyper-parameter Selection

The Adam optimizer [87] was employed for all experiments, with exponential decay of the learning rate amounting to $\gamma = 0.95$, exception made for BDD-OIA, where we added a weight-decay with parameter $\omega = 4 \cdot 10^{-5}$ and $\gamma$ was set to $0.1$, to avoid over-fitting.

The learning rate for all experiments was tuned by searching over the range $10^{-4} \div 10^{-2}$, with a total of 5 log steps. We found the SL penalty of 2 and 10 for XOR and MNIST-AddMul, respectively, to work well in our experiments. The strength of the single mitigation strategies ($\eta$) for each method was chosen accordingly to a grid-search over $\eta \in \{0.1, 0.5, 1, 2, 5, 10\}$, varying the learning rate. The best hyper-parameters were selected, in the first step, based on the highest performances in $F_1$-score for label accuracy on the validation set and, in the second step, on the lowest mitigation loss, accordingly to the value in the validation set. In particular, for SL we chose those runs that yielded the best trade-off between SL minimization and label prediction. For LTN, we found that concept supervision did interfere with the original training objective, for which the best weight for the mitigation strength was found at $\gamma = 10^{-2}$. It was adopted for both LTN+C and LTN+R +C in **Q2** of Section 6. The hyper-parameters for the combined mitigation strategies were selected according to the aforementioned criterion, by only searching the best-combined mitigation strength, while keeping the individual best strengths fixed from the previous grid search.

For BDD-OIA, we selected the learning rate ranging in the interval $10^{-4} \div 10^{-2}$ upon selecting those runs with best $F_1$-scores on labels in the validation set. The strength of the concept supervision and entropy regularization were varied in between $\{0.1, 1, 5\}$.

All best hyper-parameters for our tests are reported in the code in the Supplementary Material.

## C.4 Architectures

XOR: For this data set, we adopted two MLPs, one for the encoder $p_\theta(\mathbf{C} \mid \mathbf{G})$ and one for the decoder $p_\psi(\mathbf{G} \mid \mathbf{C})$, both with a hidden size of $3$ and ReLU activations. *This architecture is used to empirically validate RSs without forcing disentanglement.* For the disentangled case, we considered a linear layer with weight $\omega$ and bias $b$.

For SL only, we added an additional MLP, with a hidden size of $3$ and $\mathrm{tanh}$ activations, implementing the map from the logits of $\mathbf{C}$ to $\mathbf{Y}$.

MNIST-AddMul: We report here the architectures that has been used for `MNIST-Addition`, `MNIST-Multiplication`, and `MNIST-EvenOdd`. For the joint prediction, *i.e.*, without *disentanglement*, we used the encoder in Table 6. When considering *disentanglement*, we processed each digit with the encoder in Table 7 and then stacked the two concepts together. For the reconstruction, we used the decoder in Table 8. For SL only, we added an MLP with a hidden size of $50$, taking as input the logits of both concepts and processing them to the label.

Table 6: Double digit encoder for `MNIST-Addition`

| INPUT SHAPE | LAYER TYPE | PARAMETERS | ACTIVATION |
|---|---|---|---|
| $(28, 56, 1)$ | Convolution | depth=32, kernel=4, stride=2, padding=1 | ReLU |
| $(32, 14, 28)$ | Dropout | $p = 0.5$ | |
| $(32, 14, 28)$ | Convolution | depth=64, kernel=4, stride=2, padding=1 | ReLU |
| $(64, 7, 14)$ | Dropout | $p = 0.5$ | |
| $(64, 7, 14)$ | Convolution | depth=128, kernel=4, stride=2, padding=1 | ReLU |
| $(128, 3, 7)$ | Flatten | | |
| $(2688)$ | Linear | dim=20, bias = True | |

Table 7: Single digit Encoder for `MNIST-AddMul`

| INPUT SHAPE | LAYER TYPE | PARAMETERS | ACTIVATION |
|---|---|---|---|
| $(28, 28, 1)$ | Convolution | depth=64, kernel=4, stride=2, padding=1 | ReLU |
| $(14, 14, 64)$ | Dropout | $p = 0.5$ | |
| $(14, 14, 64)$ | Convolution | depth=128, kernel=4, stride=2, padding=1 | ReLU |
| $(7, 7, 128)$ | Dropout | $p = 0.5$ | |
| $(7, 7, 128)$ | Convolution | depth=256, kernel=4, stride=2, padding=1 | ReLU |
| $(3, 3, 256)$ | Flatten | | |
| $(2304)$ | Linear | dim=10, bias = True | |

Table 8: Decoder for `MNIST-Addition`

| INPUT SHAPE | LAYER TYPE | PARAMETERS | ACTIVATION |
|---|---|---|---|
| $(40, 1)$ | Unflatten | | |
| $(128, 3, 7)$ | ConvTranspose2d | depth=64, kernel=$(5, 4)$, stride=2, padding=1 | ReLU |
| $(64, 7, 14)$ | Dropout | $p = 0.5$ | |
| $(128, 3, 7)$ | ConvTranspose2d | depth=32, kernel=$(4, 4)$, stride=2, padding=1 | ReLU |
| $(32, 14, 28)$ | Dropout | $p = 0.5$ | |
| $(128, 3, 7)$ | ConvTranspose2d | depth=1, kernel=$(4, 4)$, stride=2, padding=1 | Sigmoid |

BDD-OIA: Images of `BDD-OIA` are preprocessed following [55] with a Faster-RCNN [88] pre-trained on MS-COCO and fine-tuned on BDD-100k [54]. Successively, we adopted the pre-trained convolutional layer on [55] to extract linear features, with dimension $2048$. These are the inputs for the NeSy model, which is implemented with a fully-connected NN, see Table 9.

Table 9: Fully connected layer for BDD-OIA

| INPUT SHAPE | LAYER TYPE | PARAMETERS | ACTIVATION |
|---|---|---|---|
| (2048, 1) | Linear + BatchNorm1d | dim=1024 | Softplus |
| (1024) | Linear + BatchNorm1d | dim=512 | Softplus |
| (512) | Linear + BatchNorm1d | dim=256 | Softplus |
| (256) | Linear + BatchNorm1d | dim=128 | Softplus |
| (128) | Linear + BatchNorm1d | dim=21 | Softplus |

## C.5 Confusion Matrices

Following, we report the label and concept-level confusion matrices (CMs) for XOR, MNIST-EvenOdd, and MNIST-Multiplication. For all of them, we report those obtained in runs with maximal $F_1$-score over the labels.

## C.6 XOR

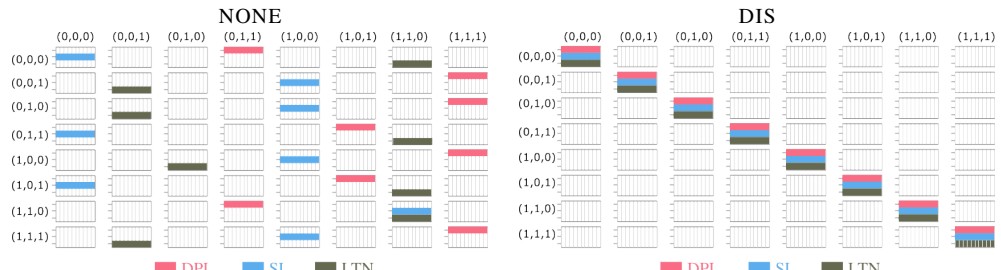

Figure 3: **CMs for** XOR: (*Left*) All NeSy models fail for RSs without any mitigation. (*Right*) Providing DIS avoids all RSs.

## C.7 MNIST-EvenOdd

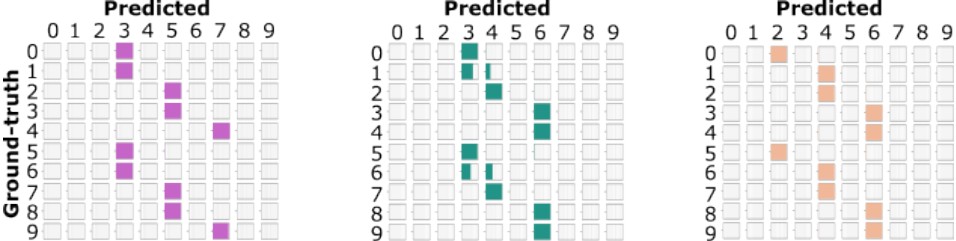

Figure 4: **NeSy models without mitigation strategies.** (*Left*) DPL picks a RS that uses only 3 digits. (*Middle*) SL optimizes for label predictions but does not always predict a correct configuration for the digits. (*Right*) LTN also picks a RS using only 3 digits.

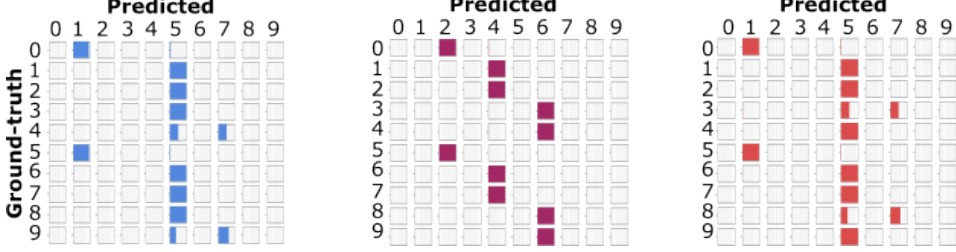

Figure 5: **NeSy models with R.** (*Left*) DPL picks a sub-optimal RS. (*Middle*) SL optimizes for label predictions but through a RS. (*Right*) LTN also picks a sub-optimal RS. For all runs, we found that R interferes with the standard learning objective of DPL and LTN, respectively.

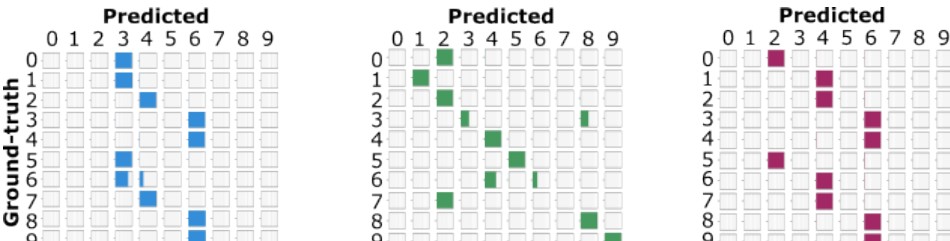

Figure 6: **NeSy models with C.** (*Left*) DPL picks a sub-optimal RS and fails to predict correctly the digits 4 and 9. (*Middle*) SL predicts correctly the 4's and 9's but does not avoid the RS. (*Right*) LTN picks a RS that uses only 3 digits.

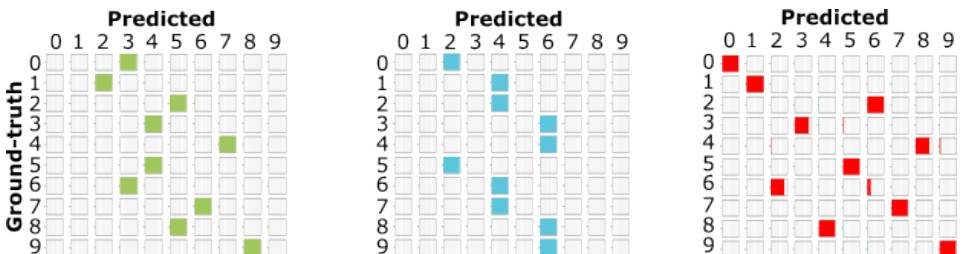

Figure 7: **NeSy models with H.** (*Left*) DPL picks a RS that uses concepts more sparsely. (*Middle*) SL picks a RS, irrespectively of the mitigation. (*Right*) LTN tends to align to the diagonal but fails to predict correctly multiple digits. The performance, nonetheless, is sub-optimal.

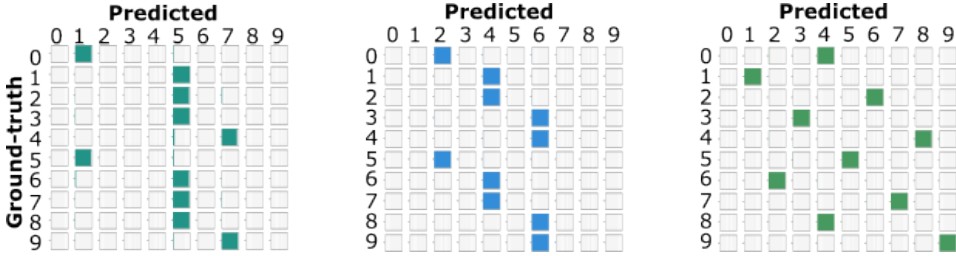

Figure 8: **NeSy models with R and H.** (*Left*) DPL picks a sub-optimal RS by using only three digits. (*Middle*) SL picks a RS. (*Right*) LTN learns correctly the odd digits but learns a RS for the even ones.

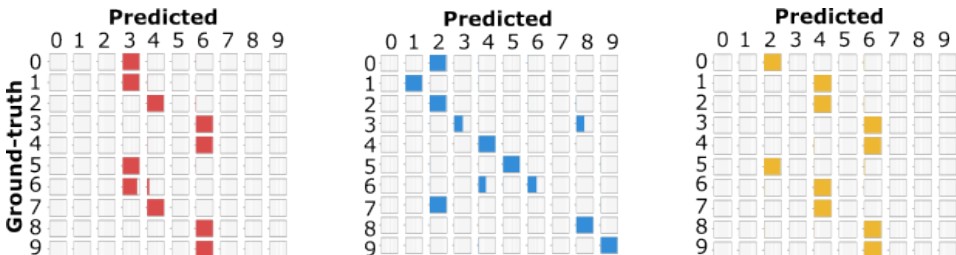

Figure 9: **NeSy models with R and C.** (*Left*) DPL picks a sub-optimal RS by using only three digits. (*Middle*) SL improves along the diagonal but fails to correctly encode four digits. (*Right*) LTN picks a RS.

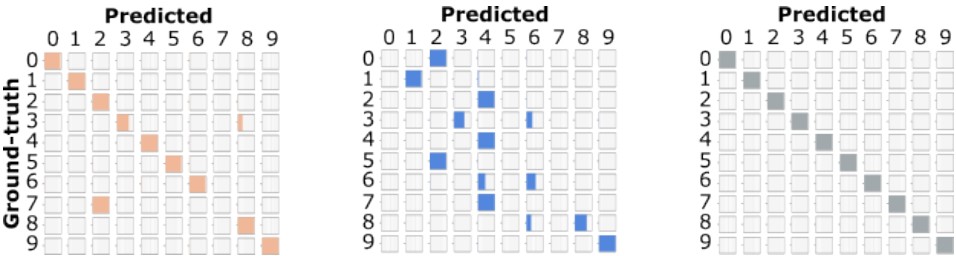

Figure 10: **NeSy models with C and H.** (*Left*) DPL correctly retrieves almost all digits but fails with the digit 7. For this method, we also found runs completely recovering the diagonal. (*Middle*) SL picks a sub-optimal RS. (*Right*) LTN avoids the RS.

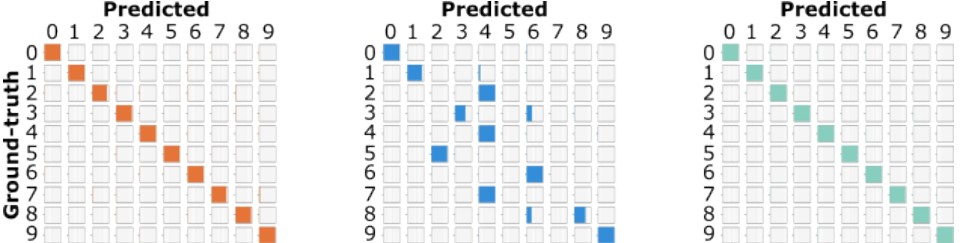

Figure 11: **NeSy models with R, C and H.** (*Left*) DPLcorrectly identifies the underlying concepts. (*Middle*) SL does not avoid RSs. (*Right*) LTN also identifies the correct digits.

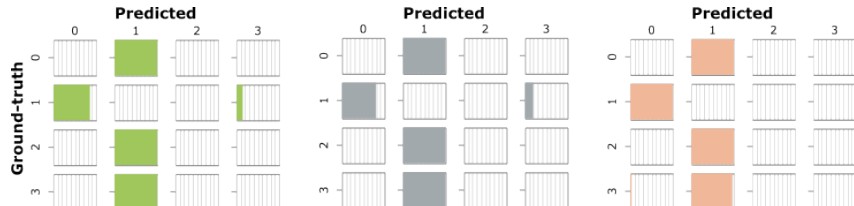

Figure 12: **CM on the addition task.** All models, DPL (*Left*), SL (*Middle*), and LTN (*Right*) fail for the RS introduced in Example 1, while failing also to predict the digit 3.

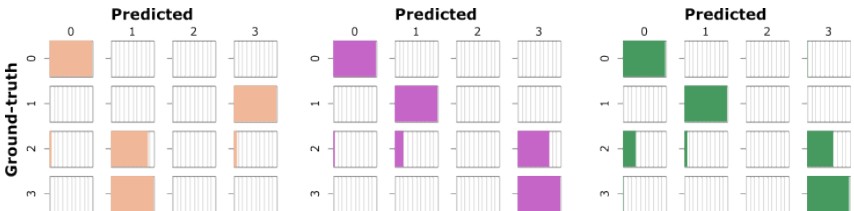

Figure 13: **CM on the product task.** (*Left*) DPL picks a RS where it fails to capture the correct semantics of all digits, except 0, (*Middle*, *Right*) SL, and likewise LTN, acquires a RS where it never predicts correctly the digit 2.

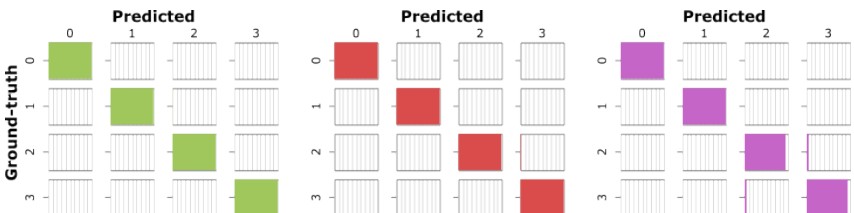

Figure 14: **CM on MTL.** Once the predictors are trained for solving addition and multiplication *jointly* through MTL, they all successfully acquire the concepts with the intended semantics: all confusion matrices are very close to being diagonal.

