| – | $85.1 \pm 4.6$ | $0.1 \pm 0.1$ | $99.3 \pm 0.2$ | $0.2 \pm 0.1$ | $98.1 \pm 0.2$ | $0.3 \pm 0.1$ |
| R | $79.8 \pm 1.0$ | $0.1 \pm 0.0$ | $99.5 \pm 0.2$ | $0.1 \pm 0.0$ | $76.3 \pm 1.1$ | $0.0 \pm 0.0$ |
| H | $98.1 \pm 0.1$ | $0.1 \pm 0.1$ | $99.4 \pm 0.1$ | $0.1 \pm 0.0$ | $81.9 \pm 0.5$ | $53.9 \pm 0.7$ |
| C | $84.9 \pm 0.1$ | $0.1 \pm 0.1$ | $99.3 \pm 0.4$ | $21.5 \pm 6.2$ | $98.1 \pm 0.2$ | $0.2 \pm 0.1$ |
| R + H | $75.4 \pm 0.4$ | $0.2 \pm 0.1$ | $99.6 \pm 0.1$ | $0.1 \pm 0.0$ | $97.9 \pm 2.3$ | $38.1 \pm 16.7$ |
| R + C | $84.0 \pm 2.2$ | $1.9 \pm 4.4$ | $99.3 \pm 0.2$ | $61.5 \pm 7.8$ | $98.1 \pm 0.2$ | $0.2 \pm 0.1$ |
| H + C | $91.9 \pm 3.5$ | $88.0 \pm 6.3$ | $99.4 \pm 0.2$ | $41.5 \pm 8.2$ | $98.2 \pm 0.2$ | $98.6 \pm 0.1$ |
| R + H + C | $95.4 \pm 0.4$ | $96.2 \pm 0.2$ | $99.5 \pm 0.2$ | $47.2 \pm 9.8$ | $98.1 \pm 0.3$ | $98.5 \pm 0.2$ |

| MNIST-AddMul | DPL | | SL | | LTN | |
|---|---|---|---|---|---|---|
| | $F_1$ (**Y**) | $F_1$ (**C**) | $F_1$ (**Y**) | $F_1$ (**C**) | $F_1$ (**Y**) | $F_1$ (**C**) |
| ADD | $68.1 \pm 6.7$ | $0.0 \pm 0.0$ | $99.5 \pm 0.2$ | $0.0 \pm 0.1$ | $67.4 \pm 0.1$ | $0.0 \pm 0.0$ |
| MULT | $100.0 \pm 0.0$ | $37.6 \pm 0.2$ | $100.0 \pm 0.0$ | $76.1 \pm 11.7$ | $98.1 \pm 0.5$ | $78.1 \pm 0.4$ |
| MULTIOP | $100.0 \pm 0.0$ | $99.8 \pm 0.1$ | $100.0 \pm 0.0$ | $99.8 \pm 0.1$ | $98.3 \pm 0.2$ | $98.3 \pm 0.2$ |