# OpenReview forum: "Not All Neuro-Symbolic Concepts Are Created Equal: Analysis and Mitigation of Reasoning Shortcuts"
_NeurIPS.cc/2023/Conference — NeurIPS 2023 poster_

### Official Review · Reviewer_p1H9 · 2023-06-29

**Soundness:** 3 good
**Presentation:** 3 good
**Contribution:** 3 good
**Rating:** 7
**Confidence:** 4

**Summary:**

This paper studies reasoning shortcuts (RS) in neuro-symbolic learning. RS refers to that the neural network does not learn a generalizable concept. This work provides a systematic characterization of RS, which not only gives a formal definition of RS, but also identifies key conditions of its occurrence. Several mitigation strategies are empirically or theoretically analyzed.

**Strengths:**

- The paper is well-motivated and well-written. I particularly appreciate the illustrative example in Figure 1.
- The definition of RS is derived from causal representation learning, and it is quite reasonable.
- Some promising theoretical results are presented, with proofs provided in the appendix.

**Weaknesses:**

- The paper focuses solely on probabilistic logic approaches like DPL, which limits its scope to a relatively narrow scope of neuro-symbolic learning.
- This paper introduces too many concepts that may be unnecessary, such as NeSy predictors, unintended semantics, concept extractors, concept distributions, deterministic distributions, etc. Therefore, it may not be easily understood by readers who are unfamiliar with this subfield.
- Some more experiments on other baseline methods are needed.



**Questions:**

- The discussed neuro-symbolic learning is essentially in a weakly supervised learning setting (correct me if I'm wrong) [1]. Therefore, can the RS and its analysis be extended to general supervised learning problems?
- From my understanding, using Shannon entropy loss is critical to avoid shortcuts. As mentioned in the last paragraph of section 5.4, entropy loss conflicts with NeSy prediction, but [1] proposes using an annealing strategy to gradually remove this entropy regularization. Furthermore, there are quite a few classic methods using maximal entropy-like regularization, e.g., mixup [2], label smoothing [3], focal loss [4], energy-based models [5], and Logit normalization [6]. I strongly recommend separating this part as an additional subsection to reveal this clue for alleviating RS.

[1] Qing Li, Siyuan Huang, Yining Hong, Yixin Chen, Ying Nian Wu, Song-Chun Zhu. Closed Loop Neural-Symbolic Learning via Integrating Neural Perception, Grammar Parsing, and Symbolic Reasoning.

[1] Zenan Li, Yuan Yao, Taolue Chen, Jingwei Xu, Chun Cao, Xiaoxing Ma, Jian Lu. Softened Symbol Grounding for Neuro-symbolic Systems.

[2] Luigi Carratino, Moustapha Cissé, Rodolphe Jenatton, Jean-Philippe Vert. On Mixup Regularization.

[3] Rafael Müller, Simon Kornblith, Geoffrey Hinton. When Does Label Smoothing Help?

[4] Jishnu Mukhoti, Viveka Kulharia, Amartya Sanyal, Stuart Golodetz, Philip Torr, Puneet Dokania. Calibrating Deep Neural Networks using Focal Loss.

[5] Connor Pryor, Charles Dickens, Eriq Augustine, Alon Albalak, William Wang, Lise Getoor. NeuPSL: Neural Probabilistic Soft Logic.

[6] Hongxin Wei, Renchunzi Xie, Hao Cheng, Lei Feng, Bo An, Yixuan Li. Mitigating Neural Network Overconfidence with Logit Normalization.

**Limitations:**

No, the limitations of the method were not discussed. The work does not have negative social impacts.

---

> ### Author Rebuttal · Authors · 2023-08-09
>
> We thank the reviewer for the positive comments about our work, in particular for finding it well-motivated and well-written. Below, we address the reviewer’s concerns.
>
> **Evaluated approaches are not exhaustive**
>
> We study reasoning shortcuts (RSs) in NeSy problems where knowledge is explicit and provided upfront.  This setting is very common in NeSy AI, as shown by the list of references we provided in the ``NeSy predictors’’ paragraph (other references were cut due to space constraints).  Within this setting, DPL, SL, and LTN are very representative.  DPL and SL are based on probabilistic logic: DPL introduces a specific layer on top of the concepts to perform logic combination, while SL uses a regularization loss to integrate logic. LTN is a representative of fuzzy logic, whereby it evaluates the satisfiability of the logical formula via fuzzy relaxation of the operators.
>
> We completely agree that RSs in settings where the knowledge is learned are equally interesting, and we do plan to look into them in a follow-up work.
>
> While we do conjecture RSs to be a widespread phenomenon, as we noted in the Related Work section (page 8), NeSy approaches are far too heterogeneous to possibly experiment with them all in any given paper.  We plan on extending our evaluation in a longer version of the paper.
>
> **Notation heavy:**
> We agree that the paper introduces several concepts, however, they are necessary to properly formalize our learning problem, even more so given this paper aims to lay the theoretical foundations of reasoning shortcuts.
>
> **Supervised vs weakly supervised**
>
> We agree that our learning setup can be seen as a weakly-supervised setting according to the definition in [1]. That is (latent) concepts $C$ are learned with supervision on labels $Y$ and leveraging knowledge $K$.
> We are not sure what the reviewer means by general supervised learning. Supervising also the concepts $C$ can be seen as a very effective mitigation strategy (concept supervision in the text) for which the count is reduced by the constraint in Prop. 5.
> On the contrary, if the reviewer means regular neural classifiers in which the label $Y$ is predicted from the input $X$ without computing concepts $C$ explicitly, our results can be adapted when knowledge $K$ is used among labels. In fact, in this case, SL can be adapted to regularize the predictions $Y$ with the known logical constraints. If some of the labels are latent, reasoning shortcuts can appear, though, in a different form. We believe this constitutes an interesting extension of our present work.
>
> **Shannon Entropy**
>
> Thank you for the references.
> These works focus on increasing the entropy of $p(Y | X)$.  This is insufficient to prevent reasoning shortcuts, which affect $p(C | X)$.  While it is possible to adapt these strategies - for instance, label smoothing - to $p(C | X)$, doing so might conflict with the objective of recovering the ground-truth concept distributions, that is, the ``correct semantics’’.  This is because, in general, there is no guarantee the ground-truth concepts themselves are high-entropy.  We will cite the provided works and discuss this point in our paper.  We will also convert the paragraph to a subsection as you suggested.
>
> Please note that we did experiment with a strategy encouraging high entropy on $p(C)$, taken from [Manhaeve et al., 2021] and denoted +H in our tables.  As noted in the ``Other heuristics’’ paragraph on page 7, Shannon Entropy can be useful in practice to regularize the network’s prediction, but its learning objective can conflict with the original learning loss (e.g. maximum log-likelihood for DPL).  However, doing so means that it is no longer possible to explicitly count the number of deterministic RSs as in Eq. (6). Notice that imposing a Shannon entropy on the concepts $C$ is equivalent to minimizing the Jensen-Shannon divergence between the concept distribution $p(C)$ and the uniform distribution.  Since the data is often not distributed uniformly, optimizing the JS divergence conflicts with recovering the true concept distribution (especially if there are few concepts, like in our experiment Q2 bottom).  In practice, the +H strategy can help to prevent some RSs, but it is insufficient to avoid all of them, as shown in Table 3, top.

---

> > ### Comment · Reviewer_p1H9 · 2023-08-15
> > **Reply**
> >
> > Thank you for clarifying my concerns. I would like to further discuss the Shannon entropy regularization. I agree that using a Shannon entropy with a constant coefficient may cause conflicts. However, this issue can be properly addressed by applying an adaptive strategy, such as focal loss [4] or annealing strategy [1].
> >
> > In general, this is a good paper, I have raised my score to 7.

---

### Official Review · Reviewer_Qpe9 · 2023-07-04

**Soundness:** 4 excellent
**Presentation:** 4 excellent
**Contribution:** 3 good
**Rating:** 7
**Confidence:** 3

**Summary:**

This paper proposes mitigation and evaluation strategies for "reasoning shortcuts" in neuro-symbolic reasoning models, roughly corresponding to when the concepts (i.e., latent variables) extracted do not match the "true" ground-truth factors. The paper then examines reasoning shortcuts on several proposed, simple datasets, including XOR, MNIST-Addition, MNIST-EvenOdd, MNIST-AddMul, and BDD-OIA.

**Strengths:**

The topic is important and underexplored, while the approach taken in this paper is novel. In general, the presentation is also excellent. The experiments, while fairly simple, are appropriate given the topic. The work is highly related to and extends prior work on disentanglement in machine learning, which is referenced (though not especially explicitly discussed). It is also decently well-motivated and some of the tasks like the BDD dataset highlighted interesting failure cases.

**Weaknesses:**

Fundamentally, this paper is subject to many of the same challenges as prior work on disentanglement. That is, the task of identifying the "correct" latent variables is the task of disentanglement. While Locatello et al. (2019)'s "Challenging common assumptions" is cited for the definition of disentanglement (as [44]), the key lesson is never explicitly referenced (at least, not attributed): without supervision or inductive biases on the model and data, disentanglement is impossible. However, the authors highlight a similar point empirically, that "Disentanglement is not enough under selection bias." Of course, this is not a particularly high bar, and indeed, the strategies taken to mitigate reasoning shortcuts in this paper do so with varying degrees of explicitness. For example, the paper's "data-based mitigation" explicitly provides supervision using a subset of the latent variables. However, in their objective-based approach, which draws inspiration from autoencoders, they could have plausibly also built on disentanglement-focused literature like InfoGAN (Chen et al. 2016) or $\beta$-TCVAE (Chen et al. 2018). However, with that being said, it approaches many of these questions from a refreshing new perspective.

This leads to the second concern - to some extent, many of the proposed mitigation strategies are standard in the disentanglement literature. For example, "On the relationship between disentanglement and multi-task learning" (Maziarka et al. 2021) highlights the role of training on multiple tasks for learning disentangled representations -- in this paper, that is seemingly equivalent to knowledge-based mitigation. On the other hand, the data-based mitigation seemingly corresponds to the approach taken in "Semi-Supervised StyleGAN for Disentanglement Learning" (Nie et al. 2020). The objective-based mitigation (i.e., reconstruction loss) is, of course, central to many representation learning approaches. Then in the discussion of "architecture-based mitigation," the connection to disentanglement work is made explicit. However, approaching these questions from the perspective of reasoning is fairly novel and is an important connection.

Ultimately, while the problem itself is well-motivated and novel, and the paper itself is excellent, and I believe it should likely be accepted based on the analysis alone, my primary reservation stems from the question of how reasoning, in particular, has shaped the mitigation strategies taken in this paper, beyond the datasets considered.

---- Post-rebuttal update ----
Most of my concerns have been addressed - I've raised my score to a 7.

**Questions:**

What differences would you highlight between the reasoning-focused tasks in this paper and the more traditional tasks in the disentanglement literature? How are the approaches discussed in this paper particularly well-suited for those tasks?

**Limitations:**

I think some more explicit discussion of the theoretical limitations demonstrated by related works (e.g., Locatello et al. 2019) would have been useful.

---

> ### Author Rebuttal · Authors · 2023-08-09
>
> We thank the reviewer for the positive comments about our work, in particular for finding it excellent, novel, and well-presented. Below, we address the reviewer’s concerns.
>
> **Disentanglement vs. RSs**
>
> Thank you for bringing up this interesting connection. There is a strong link between RSs and the problem of identifiability of discrete latent variables [1].  Identifiability is however different from (and stronger than) disentanglement.
>
> Notice also that, whereas most works on identifiability seek (im)possibility conditions, we go one step further and provide a count of the number of Reasoning Shortcuts, i.e., ways in which the model can fail to identify the right concepts.
> Hence, we are not addressing conditions to achieve identifiability but, rather, we describe the causes for non-identifiability.
>
> As for specific differences, notice that research on disentanglement focuses on continuous latent factors (e.g., in non-linear independent component analysis and disentangled representation learning via VAEs), does not consider symbolic knowledge, and typically allows factors to be permuted.  In contrast, in our work, we deal with discrete concepts learned from prior knowledge and do not allow permutations of factors.
>
> We completely agree that techniques from causal representation learning provide useful tools for understanding neuro-symbolic integration, and we hope our work serves as a starting point for this research direction.
>
> **Strategies are not novel**
>
> Our paper aims to bring the issue of RSs - and their impact on trustworthiness - to the spotlight and to understand the root causes thereof.
>
> As mentioned in the abstract, we study a set of **natural** mitigation strategies, which however only become obvious given our identified root causes.  We do not claim these are novel or technically advanced.  Rather, our contribution is making the impact they have on the number of reasoning shortcuts explicit (cf. Table 1).
>
> We are aware that (weak) supervision has been used to encourage disentanglement.  The work of Maziarka et al. is however new to us;  we will reference it in the MTL section, thank you for the pointer.  However, these works seek disentanglement, while we are concerned with a stronger property, namely identifiability.
>
> Notice also that to estimate the effect of disentanglement it does not require to specify how it is achieved.  In fact, disentanglement on XOR and MNIST datasets can be enforced architecturally and tested empirically. In sections Q1 and Q2, we empirically investigate this point.  Regardless, the approaches indicated by the reviewer may affect different root causes and could in principle lead to a reduction of RSs.  We will include the suggested works in the discussion on how to achieve disentanglement in practice in Section 5.4.
>
> [1] Hyvärinen et al. "Identifiability of latent-variable and structural-equation models: from linear to nonlinear." arXiv:2302.02672 (2023).

---

> > ### Comment · Reviewer_Qpe9 · 2023-08-11
> >
> > Thanks for the response!
> >
> > > **Disentanglement vs. RSs**
> >
> > > As for specific differences, notice that research on disentanglement focuses on continuous latent factors (e.g., in non-linear independent component analysis and disentangled representation learning via VAEs), does not consider symbolic knowledge, and typically allows factors to be permuted. In contrast, in our work, we deal with discrete concepts learned from prior knowledge and do not allow permutations of factors.
> >
> > I do not believe this fully matches the use of disentanglement in the literature. While discrete, ordered variables have not been the primary focus of disentanglement literature (arguably, out of convenience), both aspects have been explored. The question of how to learn discrete latent factors has attracted a decent amount of interest in several contexts, and one can find dozens of papers on this topic [1-4, and many others]. Moreover, most (probably all standard) disentanglement methods that provide supervision do not allow arbitrary permutation of variables (e.g. [5] mentioned in the original review and 6]).
> >
> > > **Strategies are not novel**
> >
> > As noted in the original review, I believe the framing and specific motivation in this work are novel, and in general, am positive about this work. However, given that there is a body of prior work which aims to implement similar techniques to these mitigation strategies, I believe it is worth highlighting these connections more explicitly.
> >
> > Refs:
> > 1. Learning Disentangled Joint Continuous and Discrete Representations, Dupont 2018
> > 2. Disentangling generative factors in natural language with discrete variational autoencoders, Mercatali and Freitas 2021
> > 3. Structured Disentangled Representations, Esmaeili et al 2019
> > 4. Learning Disentangled Discrete Representations, Friede et al 2023
> > 5. Semi-Supervised StyleGAN for Disentanglement Learning, Nie et al 2020
> > 6. Weakly-Supervised Disentanglement Without Compromises, Locatello et al 2020

---

> > > ### Author Response · Authors · 2023-08-12
> > > **Reply by Authors**
> > >
> > > Thank you for the quick reply and your effort in engaging with us!
> > >
> > > In light of this discussion, **we agree related work on disentanglement should be covered more thoroughly in the main text, and we plan to do so in the Related Work section**.  Please let us know if you’d prefer we address this differently.
> > >
> > > **On disentanglement.**  We stand corrected, you are right that there exists also work on disentanglement of discrete variables.
> > >
> > > We acknowledge that there exist different definitions of disentanglement in the literature.  The reason why we distinguish between disentanglement and identifiability is based on our reading of [Suter et al., Reddy et al., Hyvarinen et al.].  Our view is that factors are disentangled iff they can be manipulated (through do-operations) independently from each other [Suter et al., Reddy et al.], while identifiability amounts to finding the "correct factors’’ [Hyvarinen et al.], regardless of whether are disentangled or not.  There are also weaker notions of identifiability (such as "weak identifiability’’ [Hyvarinen and Morioka] and "alignment’’ [Marconato et al.]) that lie in-between disentanglement and identifiability. Note also that if the ground-truth factors are correlated to one another, and the learned concepts correctly identify them, then these cannot be disentangled (according to [Suter et al., Reddy et al.]).  Our analysis also covers this case, in that it makes no assumptions on the G’s being disentangled.  This is why we are keen on keeping this distinction.
> > >
> > > [Suter et al.] Suter, Miladinovic, Schoelkopf, Bauer. “Robustly disentangled causal mechanisms: Validating deep representations for interventional robustness.” ICML 2019.
> > >
> > > [Reddy et al.] Reddy, Balasubramanian, Vineeth, and others. “On causally disentangled representations”. AAAI 2022.
> > >
> > > [Hyvarinen and Morioka]. "Nonlinear ICA of temporally dependent stationary sources." Artificial Intelligence and Statistics. PMLR, 2017.
> > >
> > > [Hyvärinen et al.] "Identifiability of latent-variable and structural-equation models: from linear to nonlinear." arXiv:2302.02672 (2023).
> > >
> > > [Marconato et al.] “Glancenets: Interpretable, Leak-proof, Concept-based Models”, NeurIPS 2022.
> > >
> > >
> > > **Additional pointers to strategies for disentanglement.**  We now see better how to address this: we plan to cite relevant prior work on approaches in line with the mitigation strategies we identified in the main text.  We never intended to dismiss these works.

---

### Official Review · Reviewer_szNE · 2023-07-04

**Soundness:** 3 good
**Presentation:** 3 good
**Contribution:** 3 good
**Rating:** 6
**Confidence:** 2

**Summary:**

_Background_: The authors are interested in studying the properties of sequential, two-stage neuro-symbolic pipelines. Stage 1 is a neural network that reduces a high dimensional input (eg: an MNIST image) to a low-dimensional relaxation of a symbolic space (eg: 10 dimensional onehot vector) and Stage 2 is a probabilistic function (neural network or otherwise) that predicts the labels given the symbols utilizing a provided knowledge base. However, when trained end-to-end, the symbolic representation often collapses and the symbols learned are not equivalent to the ground truth symbols. This is called a reasoning shortcut (RS).
This work focuses on investigating the causes of such reasoning shortcuts and presenting suitable remedies for the same. The authors formally define reasoning shortcuts for NeSy architectures as symbolic mappings which allow the model to achieve a locally optimal loss, but whose semantics do not match the ground truth semantics.
The authors identify four factors to mitigate the occurrence of reasoning shortcuts: 1) additional prior knowledge,  2) additional data, 3) additional reconstruction targets in the objective function, and 4) additional inductive bias in model architecture design. They study the impact of these reasoning shortcuts mitigations on five NeSy prediction tasks with three NeSy algorithms.
The authors find that different combinations of mitigation strategies helps improve performance in different cases.

**Strengths:**

**Originality:**
-  While the concept of reasoning shortcuts was previously known, this paper is the first to offer a formal definition and analysis of reasoning shortcuts as a limitation of instantiations of neurosymbolic models.
- The mitigation strategies proposed are widely used in the neurosymbolic community, but they haven’t been connected before under the umbrella definition of reasoning shortcut mitigations. Hence, this paper brings a fresh perspective to why such strategies work well for NeSy systems.

**Quality and Clarity**
- The setup, definition, and analysis of reasoning shortcuts is well motivated and well explained.

**Significance**
- The paper identifies a significant problem with previous work and provides a formal definition of the problem and mitigation strategies to address the problem.
- I’m intrigued by the analysis of counting reasoning shortcuts because it presents a method to quantify how well a knowledge base describes a dataset / data generation process.


**Weaknesses:**

**Clarity:**
- Table 1: DIS is mentioned in the caption, but no row for it exists in the table.
- Table 2: It took me a long time to figure out that each number is the frequency of RSs for a baseline on a dataset. Please explain what the rows, columns, and values mean for each table.
-  Line 276: “when forcing disentanglement…” Disentanglement is a property of the architecture. How is the architecture modified to enforce disentanglement? The appendix doesn't mention this.
- “Roughly speaking, a concept F1 below 95% typically indicates an RS” I’ll defer to the author's judgement here because I’m not familiar with the data, but to me, a low concept F1 *on its own* signifies that the symbol extraction hasn’t been successful. Did the authors perhaps mean a low concept F1 but a high prediction F1 hints at the existence of RS's?
- Table 4 results: I'm not sure that I understand the results here. I defer to the authors, but I think this table and task demands extra scrutiny.
  - The prediction accuracy and the concept accuracy both seem low (< 80% as a ballpark figure). Is this close to the state-of-the-art performance for this dataset?
  - "concepts tend to align much closer to the diagonal..." I don't quite understand how being closer to the diagonal implies better concept generalization. The concepts aren't ordered, so an incorrect prediction, regardless of perceived distance from the correct concept, should be incorrect. Maybe I'm misunderstanding this sentence.

**Significance:**
- The method only compares against ablations of the same model. I'm concerned that this paints an incomplete picture about the relative performance of NeSy predictors compared to other models on these tasks. This lowers the significance of this architecture for the neurosymbolic community. I'd advise the authors to look into other fully neural and neurosymbolic methods that might have similar problem statements, and/or provide reasons for why the baselines are not suitable for the datasets. I've provided a couple of works I'm familiar with that are related (but not exactly equivalent) to the base algorithm studied in this paper.
  * ROAP[1]: ROAP presents an end to end algorithm for symbol grounding and symbol manipulation when external knowledge is not available, but we have diversity of tasks to instantiate a multi-task learning objective. I am reminded of ROAP because one of their tasks is similar to the MNIST digit grounding tasks presented here.
  * Concept Embedding Bottleneck models[2]: Concept embedding bottleneck models instantiate an extra layer that creates a concept embedding that's used for downstream prediction. I am reminded of CEM models because their algorithm is geared towards leveraging explainable symbolic information in real world tasks without loosing accuracy.
  * I'd also be interested in whether the authors think these models use the mitigation strategies presented for reasoning shortcuts.

Overall, I'm giving this paper a __weak accept__. The authors formalize an interesting problem which is definitely interesting to the broader neurosymbolic community. However, the related work / evaluation focuses on a very specific definition of neurosymbolic models.

[1] https://arxiv.org/abs/2206.05922
[2] https://arxiv.org/abs/2209.09056

**Questions:**

See weaknesses section

**Limitations:**

The authors have adequately addressed potential limitations.

---

> ### Author Rebuttal · Authors · 2023-08-09
>
> We thank the reviewer for their interest in our paper and for finding it clear and significant. Below we address the points raised by the review.
>
> **Typos and clarifications:**
> Thank you for pointing these out, we will update Tables 1 and 2 accordingly.
> As indicated in the main text, our architectural implementation of disentanglement is described in Appendix C - and specifically in Section C.4.  We will clarify this.
>
> The concept confusion matrices illustrate the mapping between ground-truth and learned concepts.  Specifically, rows/columns each represent a concept vector c (or g) and are sorted lexicographically (i.e., the first row is the vector (0, 0, 0), the second row is (0, 0, 1), etc.)  A model unaffected by RS would have c = g, so its confusion matrix would be the identity.  The more diagonal the confusion matrix, the better the semantics acquired by the model.  We will clarify this in the text.
>
> **The 95 \% threshold:**
> We chose a threshold of 95% on the F1-score as a proxy for non-RS solutions because the best-performing approach, DPL+H+C, achieves F1(Y) around 91% and F1(C) around 88% (see results in Table 3, top) while failing to properly recognize the digit 7 (as shown by the confusion matrix in Fig. 10, Appendix C.7). More generally, high F1(Y) and sub-par F1(C) indicate a model affected by a RS (as suggested by Eq. 4.)
>
> **BDD-OIA performance:**
> In BDD-OIA, the SotA accuracy performances are obtained with the CBM-AUC [3] model.
> This method is also below 80% F1 for labels and concepts: CBM-AUC obtains F1-mean (Y) of 70.8 $\pm$ 0.1  % and F1-mean (C) of 62.1 $\pm$ 0.1%. We will include these results in the Table of results.
>
> **Comparison with other NeSy predictors**
>
> Given how broad and heterogeneous NeSy is, we had to focus on a selection of representative architectures:  DPL is a prototypical probabilistic logic approach based on a reasoning layer; SL represents penalty-based probabilistic logic models; LTN represents penalties based on fuzzy logics (see the “NeSy Predictors” paragraph on page 3).
>
> Thank you for pointing out [1] and [2]. Notice that both architectures comply with the structure of NeSy predictors shown in Fig. 2, as they extract concepts C and process them to yield a prediction Y, and as such fall within the larger scope of our results.  The main differences are:
>
> * ROAP learns knowledge during training. From our perspective, this is not a substantial difference: even if the learned knowledge resembles the ground-truth knowledge, the model is still subject to the same RSs that the latter entails.  Worse, since the knowledge is not fixed, the model is less likely to acquire concepts with the right semantics.  On the bright side, since ROAP learns multiple tasks, it naturally leverages MTL mitigation.  We will mention this as an example usage of MTL in the knowledge mitigation section.
>
> * CEM learns both concepts and a (linear) prediction layer mapping concepts to labels.  Unlike NeSy architectures, CEMs cannot leverage prior knowledge and (like other concept-based models) require full concept supervision, cf. Section 2.
>
> We fully agree that investigating RSs for neural architectures, concept-based models, and NeSy models with learned knowledge is a critical direction of research, and we plan to pursue it in future work.  We will mention this in the conclusion.
>
> [3] Sawada and Keigo "Concept bottleneck model with additional unsupervised concepts." IEEE Access 10 (2022)

---

> > ### Comment · Reviewer_szNE · 2023-08-13
> >
> > Thanks for the clarifications! I'm keeping the score where it is right now.

---

### Official Review · Reviewer_yCW5 · 2023-07-06

**Soundness:** 3 good
**Presentation:** 4 excellent
**Contribution:** 3 good
**Rating:** 7
**Confidence:** 4

**Summary:**

The paper presents a in depth analysis of reasoning shortcuts and the impact they have of the learning process of neuro-symbolic learners.
On the ground of the performed analysis, they propose 4 different mitigation strategies to alleviate the problem:

1. Knowledge-based mitigation

2. Data-based mitigation

3. Objective-based mitigation

4. Architecture based mitigation

**Strengths:**

The paper studies an important problem and it is very well presented. It is easy to follow and the formalisation of the concept of reasoning shortcuts was needed.

The experimental analysis supports the claims.

**Weaknesses:**

I have only couple of perplexities regarding the first two mitigation strategies:

1. in the knowledge-based mitigation, the authors claim that depending on the application, collecting new knowledge might not be feasible. They then proceed in proposing Multi-task learning as a practical alternative. Here it is were I get confused:
     - how can this be more practical? In addition to collecting more knowledge you now need to collect additional labels.
     - if I understand correctly, in proposition 4 in the end the conclusion is anyway that the deterministic optimum $p_{\theta}(\mathbf{C} \mid \mathbf{G})$ of the MTL loss is a deterministic optimum of a single task whose prior knowledge is the conjunction of all the prior knowledge used in the different MTL tasks. How is it then better defining different MTL tasks than collecting more knowledge?

2. In section 5.2 the authors say that RSs can occur also when the dataset is exhaustive. This to me seems more of a problem related to not choosing the right concepts, for the task, i.e., there is not a bijective correspondence between the concepts in $\mathbf{C}$ and $\mathbf{G}$. For example, suppose we have the (very simple) task of distinguishing bird images from non-bird images, and suppose we have one concept encoding the classes "pigeon, flamingo, duck" (and we have the simple background knowledge that any of these classes implies "bird"). In this case, one mitigation strategy could be to add more supervisions on the concept (as proposed), but wouldn't it be better to actually take concepts that are "meaningful" for the task?

On the line of the last point above, I was very surprised from the amount of reasoning shortcuts learnt in the MNIST-addition task in table 2. Can you provide some examples for such RSs? Also, these results are taken on *optimal* runs. What would be the results on 30 *standard runs*?  Also, can you provide some results in terms of accuracy on both the concepts and the final outcome when using disentanglement and when not?

**Questions:**

See above

---

> ### Author Rebuttal · Authors · 2023-08-09
>
> We thank the reviewer for the positive comments about our work, in particular for finding it important, needed, and well-presented. Below, we address their concerns.
>
> **Is MTL more practical than further constraining the prior knowledge?**
>
> We agree that, compared to constraining the knowledge only, multi-task learning (MTL) is more expensive in that it requires gathering both extra knowledge and extra labels.  However, MTL is a sensible choice whenever the first option is infeasible, as we noted in Section 5.1.  It is in this sense that MTL is more practical.
>
> To see this, consider the MNIST-Addition experiment (Table 3, bottom). Here, the knowledge specifies how to perform addition and no extra constraints can be added without changing the semantics of the task. This can be side-stepped by pairing the latter to a multiplication task, and doing so completely avoids RSs.
>
> Section 5.1 currently reads “experts may not be available, or it may be impossible to constrain K without also eliminating concepts with the intended semantics”.  We agree this is confusing and will change it to “experts may not be available, or it may be impossible to further specialize the knowledge K without altering the semantics of the prediction task”.
>
> **Do Reasoning Shortcuts appear when changing the underlying concepts?**
>
> We fully agree that the structure of the knowledge K and the concepts appearing therein determine whether, and how many, RSs exist.  This follows from Theorem 2.
>
> Manipulating K and C is one way of reducing the number of RSs.  One option is to replace low-level concepts with higher-level variants.  For instance, in BDD-OIA one could compact all obstacle concepts (e.g., “pedestrian”) into a *single* “obstacle” concept.  This leads to a sensible reduction of RSs, which can be evaluated using Eq. (6).$*$
>
> The main issue is that doing so is not always possible. In the XOR problem, for instance, we cannot lump together any concepts without changing the semantics of the problem.
> Identifying cases in which concept manipulation can work, and automating said procedure, is an interesting research question worth investigating in future work.
>
> $*$ This case still obeys Theorem 2 if we replace C, G, and K with their “coarser” variants.
>
> **Extended results for Table 2**
>
> **Example of RSs for MNIST-Addition:** In section C.2.2, we described the RSs affecting the MNIST-Addition task with all digits. We will clarify this in the main text. The idea is as follows: when both digits are processed together (no disentanglement) the model can map the two digits to whatever combination of concepts gives the correct sum. As an example, all the sums $G_1 + G_2 = 0 + 2 = 1 + 1 = 2 + 0 = 2$ can be mapped to the combinations $C_1=1$ and $C_2=1$.
>
> **Optimal vs. standard runs:** The reason why we considered 30 optimal runs is because reasoning shortcuts are defined as models that achieve (near) optimal performance by leveraging unintended concepts.  The performance of standard models is not representative of this definition.
>
> Regardless, we report the label and concept accuracies of 30 such models in the rebuttal PDF for DPL, SL, and LTN on both MNIST-Addition and XOR, with and without disentanglement.  All models fare much better at prediction labels rather than concepts, as expected, indicating that even “non optimal” models suffer from RSs.  These results are compatible with those reported in the main text.
>
> *More in detail*: without disentanglement (-), all models achieve high label accuracy (Acc$(Y)$) by leveraging poor quality concepts, as shown by the low concept accuracy (Acc$(C)$).  By imposing disentanglement (DIS), the concept accuracy generally improves, indicating that models get closer to the intended semantics.
>
> On the XOR data set, some of the models learned do not reach optimality, regardless of architecture (DPL, SL, LTN).  This occurs because they sometimes remain stuck in bad local minima, with and without disentanglement.  This is reflected by the large variance observed in the XOR results. Conversely, on MNISTAddition, we obtained stable runs, yielding most of the time an optimal behavior. The only exception here is LTN without disentanglement, for which we observed more variability.

---

> > ### Comment · Reviewer_yCW5 · 2023-08-13
> >
> > All my questions have been answered. I'll keep my score at 7.

---

### Author Rebuttal · Authors · 2023-08-09

We are grateful to all reviewers for taking the time to evaluate our paper and appreciating the motivation (**p1H9**), the theoretical analysis (**Qpe9**), and the significance of our work (**yCW5**, **szNE**, **Qpe9**), as well as the quality of the presentation (**yCW5**, **szNE**, **Qpe9**, **p1H9**).

We will address the points they raised in the detailed replies.

We also attach to this comment a PDF containing additional empirical results for **Q1**.

---

### Decision · Program_Chairs · 2023-09-21

**Decision:**

Accept (poster)

**Comment:**

This paper formally defines and studies reasoning shortcuts in Neuro-Symbolic predictive models and suggest different strategies to mitigate them. Reasoning shortcuts refer to solutions that achieve the best loss on the training set but don't match ground-truth concept distribution. This means models that use reasoning shortcuts will not be able to generalize to out-of-distribution cases where the same concepts are combined differently. Authors study this issue theoretically and empirically and suggest different mitigation strategies.

I think the paper could benefit from in-depth discussions that motivates this line of work for a larger audience and connects it to other areas. For example:

- Out-of-distribution generalization: This seems to be a particular case of out-of-distribution generalization. Would be nice to discuss connections.
- Disentanglement: It looks like disentanglement helps with removing reasoning shortcuts in some of the studies cases.
- Foundation Models: It seems that large vision language models trained on massive corpuses are more immune to these issues. One can think of vision language models + chain-of-thought as a symbolic reasoner. How does your work relate to this?

As for the final decision, all reviewers voted for acceptance and the paper seems to be valuable for understanding and improving Neuro-Symbolic models so I vote for acceptance.